# Lower Bounds on Randomly Preconditioned Lasso via Robust Sparse Designs

**Jonathan A. Kelner**[*]
MIT

**Frederic Koehler**[†]
Stanford

**Raghu Meka**[‡]
UCLA

**Dhruv Rohatgi**[§]
MIT

## Abstract

Sparse linear regression with ill-conditioned Gaussian random covariates is widely believed to exhibit a statistical/computational gap, but there is surprisingly little formal evidence for this belief. Recent work has shown that, for certain covariance matrices, the broad class of Preconditioned Lasso programs provably cannot succeed on polylogarithmically sparse signals with a sublinear number of samples. However, this lower bound only holds against deterministic preconditioners, and in many contexts randomization is crucial to the success of preconditioners. We prove a stronger lower bound that rules out randomized preconditioners. For an appropriate covariance matrix, we construct a single signal distribution on which any invertibly-preconditioned Lasso program fails with high probability, unless it receives a linear number of samples. Surprisingly, at the heart of our lower bound is a new *robustness* result in compressed sensing. In particular, we study recovering a sparse signal when a few measurements can be erased adversarially. To our knowledge, this natural question has not been studied before for *sparse* measurements. We surprisingly show that standard sparse Bernoulli measurements are almost-optimally robust to adversarial erasures: if $b$ measurements are erased, then all but $O(b)$ of the coordinates of the signal are identifiable.

## 1 Introduction

Random-design sparse linear regression (SLR) is a fundamental problem in high-dimensional statistics and learning theory. The simplest formulation of this problem is the following: given independent covariates $X_1, \ldots, X_m$ drawn from an $n$-variable Gaussian distribution with zero mean and positive-definite covariance $\Sigma$, and responses $y_i = \langle X_i, w^* \rangle$ for some unknown $k$-sparse signal $w^*$ (i.e. with at most $k$ nonzero entries), the goal is to recover $w^*$. While there are more complex models (e.g. with noise or non-Gaussian distributions), we are proving *lower bounds*, so the simplified model only makes our results stronger.

Information-theoretically, for *any* positive-definite covariance $\Sigma$, it's possible to recover $w^*$ exactly from only $O(k \log n)$ samples $(X_i, y_i)$. (Again, this is a special case of more general results for the noisy setting.) However, the algorithm which achieves this sample complexity is computationally inefficient — it has time complexity $O(n^k)$. Significant effort has gone into *polynomial-time*

---

[*]kelner@mit.edu. This work was supported in part by NSF Large CCF-1565235, NSF Medium CCF-1955217, and NSF TRIPODS 1740751.

[†]fkoehler@stanford.edu. This work was supported in part by NSF CAREER Award CCF-1453261, NSF Large CCF-1565235, A. Moitra's ONR Young Investigator Award, and E. Mossel's Vannevar Bush Faculty Fellowship ONR-N00014-20-1-2826.

[‡]raghum@cs.ucla.edu. This work was supported in part by NSF CAREER Award CCF-1553605 and NSF Small CCF-2007682.

[§]drohatgi@mit.edu. This work was supported in part by an Akamai Presidential Fellowship and a U.S. DoD NDSEG Fellowship.

36th Conference on Neural Information Processing Systems (NeurIPS 2022).

sample-efficient algorithms for sparse linear regression, under as weak conditions on the covariates as possible [22, 16, 44, 15, 13, 21, 20, 14, 18, 12, 45, 53, 48, 38, 47, 11, 46]. Essentially all of these algorithms are variations on one of two fundamental methods: $\ell_1$ regularization (most notably, the Lasso program [44]), and greedy variable selection (most notably, Orthogonal Matching Pursuit [45]). Moreover, both of these methods seem to hit the same fundamental barrier: they require that $\Sigma$ is in some sense *well-conditioned*. The most basic assumption enabling efficient algorithms is a bound on the condition number $\lambda_{\max}(\Sigma)/\lambda_{\min}(\Sigma)$, and this assumption has been weakened (to e.g. the Restricted Isometry Property [15], the Compatibility Condition [48], or Weak Submodularity [18]) but not eliminated. This barrier, and the lack of algorithmic techniques which evade it, suggests that sparse linear regression with general positive-definite $\Sigma$ may exhibit a computational/statistical gap: that is, sample-efficient signal recovery may be computationally hard.

**Preconditioned Lasso.** Recent work [32] provided the first computationally-efficient and sample-efficient algorithm for a broad class of *ill-conditioned* covariance matrices $\Sigma$ (specifically, covariance matrices of Gaussian Graphical Models with low treewidth such as Gaussian time-series data). The algorithm was a generalization of Lasso via an initial *preconditioning* step; while preconditioning has been enormously impactful for efficiently solving linear systems [39], it has only recently been applied to sparse linear regression. The algorithmic results of [32] raise an obvious question: *can preconditioning close the statistical/computational gap for ill-conditioned sparse linear regression? Or is the gap inherent?*

The answer is unclear. The basic obstacle is that no "candidate" hard instances of random-design sparse linear regression are known: nearly all examples on which Lasso provably fails can be trivially fixed by preconditioning [53, 23, 17, 46, 31]. The exception is the lower bound in [32], which provides a limited negative answer to the first question: there is a covariance matrix $\Sigma$ such that any *deterministic* preconditioner fails (see Section 1.1 for a concrete statement). They showed that for each preconditioner, there exists a specific bad signal on which it fails, where the signal chosen depends heavily on the preconditioner.

This suggests the tantalizing possibility that any ill-conditioned SLR problem can be solved by applying *randomized* preconditioning. This way, the signal can no longer be chosen adversarially based on the preconditioner. Indeed, numerical linear algebra is full of situations where randomized maps are useful precisely for this reason, including the JL embedding and other dimension reduction methods, algorithms based on random metric embeddings, and the famous Nystrom method for low rank approximation and preconditioning (see e.g. [56, 51, 42]). Also, it was observed in [31, 32] that randomized preconditioners (or, roughly equivalently, trying a class of preconditioners) *can* be used to solve SLR problems in various cases where we do not currently know of a fixed deterministic preconditioner that works. Given this context, we need to ask the following question: can *randomized* preconditioning close the statistical/computational gap?

**Our result.** We give a near-complete negative answer to this revised question: for a carefully-chosen $\Sigma$ and distribution over the unknown signal $w^*$, any randomly-preconditioned Lasso program with invertible preconditioners fails (see Section 2 for a more concrete statement).

**Broader context on statistical/computational gaps.** Random-design sparse linear regression is very interesting among high-dimensional statistics problems, in that there is essentially no evidence for its conjectured statistical/computational gap; while SLR with worst-case covariates (i.e. not drawn from a distribution) has been proven computationally hard under standard worst-case complexity-theoretic assumptions [37, 57, 30, 28], no such hardness is known for random-design SLR.[5] In contrast, problems as diverse as sparse PCA [7], average-case RIP certification [55], planted dense subgraph [10], robust sparse mean estimation [8], and negative-spike sparse PCA [8] have provably statistical/computational gaps under (variants of) the Planted Clique conjecture.

Why isn't sparse linear regression on this list? One possibility is that there is no statistical/computational gap; in this case, our results imply that new algorithmic techniques beyond preconditioning are needed, and our hard instances provide a natural testing ground for new techniques.

---

[5]This is when $\Sigma$ is invertible. If $\Sigma$ is not invertible the task would be learning the concept/regression function, i.e. outputting *any* $\hat{w}$ so that $(\hat{w} - w)^T \Sigma (\hat{w} - w) = 0$. If required to output a $k$-sparse predictor and $\Sigma$ is not invertible, the problem is NP hard with infinitely many samples [29] as it's equivalent to finding a sparse solution of linear equations.

But another potential explanation is the current dearth of conjecturally-hard instances of random-design sparse linear regression. Reductions from Planted Clique (and other techniques for proving average-case lower bounds) either explicitly or implicitly involve constructing distributions over the unknown concept class which encapsulate (to some degree) the hardness of the original problem. For instance, sparse PCA is conjecturally hard even when the planted sparse direction is drawn uniformly at random from $k$-sparse vectors with nonzero entries $\pm 1/\sqrt{k}$ [36, 9]. Similarly, non-reduction based lower bounds like Statistical Query lower bounds[6] are typically derived against the uniform distribution over a finite set of concepts [27]. Thus, it is problematic that for sparse linear regression, no such distribution has even been hypothesized: i.e., a family $(\Sigma_n, \mathcal{D}_n)_{n \in \mathbb{N}}$ where $\Sigma_n$ describes an $n \times n$ covariance matrix, and $\mathcal{D}_n$ describes a distribution over $k(n)$-sparse $n$-dimensional signals, specifying a model which seems to encapsulate some of the difficulty of random-design SLR. While it would certainly be necessary that the covariance matrices $\Sigma_n$ are ill-conditioned, there are also natural families of ill-conditioned covariate distributions which lead to tractable instances (e.g. [32, 31]), so such a property is by no means sufficient.

Our main result provides a family of covariance matrices and signal distributions which is hard at least for all preconditioned Lasso algorithms (this was not accomplished by [32], because they constructed a different signal for each preconditioner). Thus, our work can be viewed as a necessary first step towards average-case computational hardness of ill-conditioned sparse linear regression.

## 1.1 The Preconditioned Lasso

Perhaps the canonical method to solve sparse linear regression problems is by solving a convex program known as the Lasso [44]. In our noiseless setting, it reduces to the basis pursuit program

$$\hat{w} \in \underset{w \in \mathbb{R}^n : Xw = y}{\operatorname{argmin}} \|w\|_1.$$

This program is well-studied, and it's known to succeed with high probability with $O(k \log n)$ samples when $\Sigma$ is well-conditioned [54], but it also is known to fail if $\Sigma$ is ill-conditioned [23, 17, 31]. In the context of ill-conditioned SLR, the class of programs known as the Preconditioned Lasso [32] is significantly more powerful. Informally, these algorithms apply some change-of-basis to "condition" the covariates before solving the basis pursuit program. Formally:

**Definition 1.** For an invertible $n \times n$ matrix $S$ (which we think of as an arbitrary function of $\Sigma$ that however cannot depend on the samples), the *$S$-preconditioned Lasso* on data $(X_i, y_i)$ applies the transformation $X_i \mapsto S^{-1} X_i$, solves the basis pursuit to get an estimate $\hat{v}$, and returns $\hat{w} := S^T \hat{v}$. This corresponds to solving the convex program

$$\hat{w} \in \underset{w \in \mathbb{R}^n : Xw = y}{\operatorname{argmin}} \|S^T w\|_1. \tag{1}$$

Preconditioned Lasso obviously generalizes the Lasso, and it works for SLR tasks where applying the Lasso directly will fail badly. For example, whenever the covariates come from autoregressive time series data (e.g. random walk or $AR(1)$) the sparse linear regression problem can be solved nearly optimally via Preconditioned Lasso, even though the covariates are likely to be heavily correlated. More generally, any covariance matrix with low-treewidth dependency structure induces an SLR model that is tractable via Preconditioned Lasso, even if the matrix is arbitrarily ill-conditioned [32]. Due to this generality, examples that are provably hard against the Preconditioned Lasso are much more difficult to obtain, and much more interesting. Concretely, prior to the present work, the only known hardness result against the Preconditioned Lasso was the following statement:[7]

**Theorem 1.1** (Informal theorem statement from [32]). *For any $n > 0$, there is a positive-definite covariance matrix $\Sigma : n \times n$ such that for any preconditioner $S$, there exists some* $\operatorname{polylog}(n)$-*sparse signal $w^*$ which $S$-preconditioned Lasso with probability $1 - o(1)$ fails to recover, when given $o(n)$ independent samples $X_i \sim N(0, \Sigma)$ and $y_i = \langle X_i, w^* \rangle$.*

---

[6]Relatedly, for realizable regression problems there is a general computationally inefficient algorithm which makes a smaller number of SQ queries [49].

[7]To be more precise, Theorem 1.1 and the definition of Preconditioned Lasso actually apply to all rectangular $n \times s$ preconditioners, not just invertible preconditioners (note that Program 1 is still defined, though it no longer corresponds to a change-of-basis). However, most algorithms applying Preconditioned Lasso in the literature use invertible preconditioners [31, 32]. Moreover, restricting to invertible preconditioners does not improve the failure probability achievable by the techniques in [32].

This provides a converse to the algorithmic results of [32], which show that for certain covariance matrices, there is a preconditioner that works for all signals. However, the limitation of Theorem 1.1 is that the hard signal depends on the preconditioner: unpacking the proof of Theorem 1.1 only gives a signal distribution on which any preconditioner fails with probability $\Omega(1/n)$. Thus, the lower bound is extremely weak if we allow for *randomization* in the choice of preconditioner, which is how preconditioners are often applied in other contexts.

## 2 Main Results

In this paper, we eliminate that limitation and provide a *single* signal distribution which is provably impossible to precondition. That is, we construct a covariance matrix and a sparse signal distribution under which Preconditioned Lasso with any invertible, randomized change-of-basis must fail *with high probability*, unless a linear number of samples are given.

**Theorem 2.1** (Theorem E.3 of the Appendix). *Let $n > 0$. There is a positive-definite covariance matrix $\Sigma : n \times n$ and a distribution $\mathcal{D}$ over $\mathrm{polylog}(n)$-sparse signals with the following property: for any invertible randomized preconditioner $S$, if we draw $w^* \sim \mathcal{D}$, then $S$-preconditioned Lasso fails to recover $w^*$ with probability at least $1 - o(1)$, when given $o(n)$ independent samples $X_i \sim N(0, \Sigma)$ and $y_i = \langle X_i, w^* \rangle$.*

In fact, the full version of Theorem E.3 is even stronger: it shows that even if we fix a family of $\mathrm{poly}(n)$ different preconditioners, they will all fail on a problem instance sampled from our distribution with probability $1 - o(1)$. This allows us to rule out an even larger class of algorithms: for example, the algorithm used in [31] for solving jointly walk-summable SLR instances adaptively selects one out of $n$ possible (invertible) preconditioners before running the Preconditioned Lasso, and our lower bound shows that this strategy and variants are provably defeated by our new construction.

Additionally, we can extend our result to show hardness against rectangular $n \times s$ preconditioners. For technical reasons we only achieve a failure probability of $1/2 - o(1)$, and require a bound on the preconditioner size. Nonetheless, at this failure probability and with poly-logarithmically sparse signals, we can rule out all polynomially-sized preconditioners.

**Theorem 2.2** (Theorem E.10 of the Appendix). *Let $n > 0$. There is a positive-definite covariance matrix $\Sigma : n \times n$ and a distribution $\mathcal{D}$ over $k$-sparse signals, for any $k \geq \log^{12}(n)$, with the following property: for any randomized preconditioner $S$ with at most $\exp(k/\log^{10}(n))$ columns, if we draw $w^* \sim \mathcal{D}$, then $S$-preconditioned Lasso fails to recover $w^*$ with probability at least $1/2 - o(1)$, when given $o(n)$ independent samples $X_i \sim N(0, \Sigma)$ and $y_i = \langle X_i, w^* \rangle$.*

In both theorems, the probability is over the choice of preconditioner, the signal distribution, and the random samples. Surprisingly, the key technique used to prove Theorem 2.1 and 2.2 is a *positive* identifiability result in compressed sensing. We show that random sparse compressive design matrices are robust to adversarial erasures. To prove our lower bounds, we will fix one such matrix $M$, and we will define the covariance matrix $\Sigma = (M^T M + \epsilon I)^{-1}$ for small $\epsilon > 0$; the robustness of $M$ will then imply important properties of $\Sigma$. *Note that the final Gaussian SLR design matrix (with independent rows drawn from $N(0, \Sigma)$) will be quite different from the erasure-robust design matrix $M$.* But first, we describe the application of such matrices $M$ to compressed sensing, as we believe it is of independent interest and a main contribution of this work.

### 2.1 Key Technique: Erasure-Robust Sparse Designs

In this section we describe the key technique that enables our main results, an erasure-robust sparse design, and provide an independent motivation for this design from compressed sensing. At the end, we provide intuition for how it connects back to hardness against Preconditioned Lasso.

There is a vast literature on sparse linear regression and compressed sensing. Many deterministic conditions and stochastic models for the measurement matrix (also known as design matrix or covariate matrix) have been demonstrated to imply that sparse signals can be recovered either information-theoretically or algorithmically [54]. In noisy settings, the goal is usually either approximate recovery under an $\ell_p$ norm or prediction error (e.g. [14, 53]); exact support recovery with some assumptions about the signal-to-noise ratio (e.g. [53]); or approximate support recovery under distributional assumptions about the signal (e.g. [40]). However, in noiseless settings, the goal is

invariably exact recovery. This is obviously ideal. But in situations where the measurements are not entirely controlled by the algorithm designer, exact recovery could be impossible. A natural goal is then to try to recover *part* of the signal. To our knowledge, this notion of partial recovery of sparse signals (i.e. due to shortcomings of the measurement matrix rather than due to noise) has received essentially no attention; see Section 3 for a discussion of related notions.

Part of the reason may be that it's not obvious what models for a compressive measurement matrix exhibit the behavior that some but not all of the coordinates of a sparse signal are identifiable, besides artificial examples where e.g. unconstrained variables are added to the system. Such examples do not answer the question of whether partial recovery is possible under fundamentally weaker modelling assumptions than total recovery.

**Erasure-robustness.** Our key technical contribution is a proof that partial recovery is possible in a natural *semi-random* model. Specifically, we show that random sparse compressive measurement matrices $M$ are "erasure-robust", by which we mean that if an adversary *erases* a small fraction of the measurements $\langle M_i, x^* \rangle$, replacing them with question-marks, then most of the coordinates of the sparse signal vector $x^*$ are still information-theoretically identifiable. Moreover, the identifiability result is stable under additive measurement noise with polynomially small norm.

Adversarial erasures have been studied in compressing sensing before, and of course have also been long and extensively studied in coding theory (see e.g. [35, 33, 24]). For random dense compressive matrices, it's known that deleting a small fraction of the measurements essentially does nothing; the sparse signal is still totally recoverable with high probability [19, 52, 34]. But for random sparse matrices (which in the absence of erasures do also enable total sparse recovery [5]), no such robustness has been proven, because it's not true: the adversary may simply delete all measurements interacting with a particular coordinate, rendering that coordinate unidentifiable. Given this, partial recovery is the best that can be hoped for. Our result implies that it is also attainable, at least information-theoretically. Moreover, we achieve a nearly-tight bound on the number of unidentifiable coordinates. Here is the informal statement:

**Theorem 2.3** (Theorem C.14 of the Appendix). *Let $n, m > 0$ satisfy $n > m > \Theta(\log^2 n)$, and let $M$ be an $m \times n$ matrix with independent Bernoulli-$p$ entries for $p = \Theta(\log^2 n)/m$. With high probability, the following holds. For any set of "deleted" equations $B \subseteq [m]$ of size $|B| \leq O(m/\operatorname{polylog}(n))$, there is a set $C$ (the "unidentifiable coordinates") of size $|C| \leq 2|B|$ such that*

$$\|x_{C^c}\|_2 \leq \operatorname{poly}(n) \cdot \|M_{B^c}x\|_\infty$$

*for any $O(m/\operatorname{polylog}(n))$-sparse vector $x \in \mathbb{R}^n$.*

To make apparent the implication for compressed sensing, we state the following corollary:

**Corollary 2.4.** *Let $M \in \{0,1\}^{m \times n}$ be a fixed, known matrix for which the guarantee of Theorem 2.3 holds. Let $x^* \in \mathbb{R}^n$ be an unknown $k$-sparse vector with $k \leq O(m/\operatorname{polylog}(n))$. Define measurements $y = Mx^* + \eta$ for some noise vector $\eta$ with $\|\eta\|_\infty \leq \delta$, but suppose that $y$ undergoes adversarial erasures. That is, we only observe $\overline{y} \in \mathbb{R}^m$ where*

$$\overline{y}_i = \begin{cases} ? & \text{if } i \in B \\ y_i & \text{if } i \in B^c \end{cases},$$

*for an adversarially-chosen set $B \subseteq [m]$. Then we can information-theoretically identify an **estimate** $\hat{x} \in \mathbb{R}^n$ and a **region of uncertainty** $C \subseteq [n]$ such that $|\hat{x}_i - x_i| \leq \delta \cdot \operatorname{poly}(n)$ for all $i \notin C$. Moreover, the region of uncertainty is a function of $B$ (not depending on $x$ or $y$) and is bounded as $|C| \leq 2|B|$.*

In particular, the estimator is simply $\hat{x} = \operatorname{argmin}_{x:\|M_{B^c}x - \overline{y}_{B^c}\|_\infty \leq \delta} \|x\|_0$. The region of uncertainty $C$ is any set $C$ satisfying the guarantee of Theorem 2.3 with respect to $B$; it can be (inefficiently) computed from $B$ via brute-force computation of singular values of submatrices of $M$. Then, since $\hat{x} - x^*$ is $2k$ sparse, the guarantee of Theorem 2.3 gives that

$$\begin{aligned} \|(\hat{x} - x^*)_{C^c}\|_\infty &\leq \operatorname{poly}(n) \cdot \|M_{B^c}(\hat{x} - x^*)\|_\infty \\ &\leq \operatorname{poly}(n) \cdot \|(M_{B^c}\hat{x} - \overline{y}_{B^c}) + (\overline{y}_{B^c} - M_{B^c}x^*)\|_\infty \leq \operatorname{poly}(n) \cdot \delta \end{aligned}$$

as claimed in the corollary.

Sparsity of the measurement matrix (and not just the signal) is well-studied in compressed sensing and has various practical applications. For example, in scientific experiments it is often the case that linearity of the response with respect to the covariates is a modelling assumption that's only reasonable for a sparse covariate vector [26]. Our result implies that even with a sparse measurement matrix, adversarial erasures (due to e.g. experimental error) are not disastrous. (Note that in the above theorem, each row of the measurement matrix is roughly $(n \log^2 n)/m$ sparse, which up to logarithmic factors cannot be improved, even without erasures, since the measurement matrix must have $\Omega(n)$ nonzero entries).

*Open question.* Our results show that partial sparse recovery in this semi-random model is possible. However, finding a computationally efficient algorithm is an interesting open problem.

**Connection to lower bounds against Preconditioned Lasso.** It may seem rather mysterious that construction of an erasure-robust, "good" design matrix is the key ingredient in a distributional hard example for a family of sparse recovery algorithms. The technical reasons for this connection are deferred to the overview, but here we try to give some high-level intuition. First, for intuition, we restrict our focus to sparse preconditioners, because dense preconditioners (morally) do not preserve the sparsity of the unknown signal and therefore should not work. Now, if $\Sigma$ is very ill-conditioned, the preconditioner essentially needs to "fix" $\Sigma$ by reweighting the different eigenspaces.

If we are allowed to construct the signal based on the preconditioner, then the preconditioner is forced to be a good approximation for $\Sigma$ everywhere, with no bad directions. But from any good measurement matrix, using the fact that it has dense kernel, we can construct a $\Sigma$ so that no sparse preconditioner can approximate $\Sigma$ everywhere. This is the approach taken in [32] for Theorem 1.1.

In our case, we need to construct the signal distribution without knowing the preconditioner, so the preconditioner is only forced to be a good approximation for $\Sigma$ in *most* directions. Ruling out sparse preconditioners then corresponds to a measurement matrix which has a density property even if some of the rows are ignored (specifically, erasure-robustness). Sparsity of the measurement matrix is needed so that the rows are valid sparse signals, and compressivity is needed so that $\Sigma$ is ill-conditioned in many directions.

## 3 Related Work

**Hard Examples for SLR.** As we mentioned earlier, SLR with worst-case covariates is known to be computationally hard under worst-case complexity-theoretic assumptions [37, 57, 30, 28]. However, for the random-design covariate model, there is no known reduction-based hardness, even under average-case or cryptographic assumptions. We enumerate known restricted hardness results:

First, there is a large literature on when the Lasso and Basis Pursuit programs fail at sparse recovery, even for random designs [53, 23, 17, 46, 31]. While these results do (technically speaking) prove lower bounds against classes of algorithms, these classes are quite small; the constructed examples are only proven to be hard for the Lasso and/or Basis Pursuit (at best, these programs have one meta-parameter). In fact, as has been previously observed [58, 32], all of the "hard" examples provided in the above works can be fixed by a simple change-of-basis.

Second, there is a more general lower bound against the class of convex programs solving least-squares regression with coordinate-separable regularizers [58]. While this is a fairly broad class of algorithms (incomparable with the Preconditioned Lasso), the result has two limitations. One is that the constructed hard signals depend on the regularizer, so there is no single signal distribution that is hard for the entire class. The other limitation is that, like in the previous works on hardness against the Lasso, the hard example in [58] can be made easy for the Lasso by a simple change-of-basis.

Third, in [32], motivated by the latter limitation, covariance matrices are constructed such that for any change-of-basis, there is a sparse signal (in the original basis) which causes the "preconditioned" basis pursuit program to fail. However, as we have previously noted, this result is still limited by the strong dependence of the signal on the preconditioner: it does not even rule out the possibility that there are always *two* preconditioners so that every signal can be recovered by one of them.

Fourth, for *isotropic* random-design SLR (i.e. when $\Sigma = I$), there has been work on identifying the precise sample complexity of sparse recovery. In particular, there appears to be a constant-factor gap between the sample complexities of algorithmic recovery and information-theoretic recovery.

Evidence has been given for this gap via the Overlap Gap Property [25], which implies the failure of a restricted class of "stable" algorithms. However, this problem seems fundamentally different from the problem we consider, where the hardness arises from the ill-conditioning of the covariates, and the sample complexity gap is conjecturally exponential rather than a constant.

**Partial sparse recovery.** There are several other works in compressed sensing that use the terminology of "partial" recovery. To our knowledge, these works all consider different settings from ours; we explain the differences. First, in [3], partial sparse recovery refers to totally recovering a signal that is only partially sparse (where the signal space is divided into two sets of coordinates, and it's known that the signal is sparse on the first set). Second, in [43], the goal is indeed to recover only part of the support of the signal. However, their model is the Gaussian Sequence Model (i.e. where the measurement matrix is the identity), where it is obvious that partial recovery is possible, because there is no compression. Third, as discussed previously, one common goal in noisy models is partial support recovery (see e.g. [40]). There, the goal is to estimate the support with few false positives and false negatives, and the reason for error is simply that some coordinates of the signal may be very small and therefore indistinguishable from noise. In contrast, partial identifiability occurs in our setting even without noise, due to a weaker model for the measurement matrix. Moreover, proving partial support recovery in the setting of [40] requires make strong probabilistic assumptions about the signal, e.g. that the support is a uniform sparse set. In contrast, our results prove conditions under which a measurement matrix enables partial recovery of arbitrary sparse signals.

# 4   Technical Overview

We start with a sketch of the proof of Theorem 1.1 from [32], which only achieves a failure probability of $O(1/n)$, and which formally motivates the need for erasure-robust sparse designs. We then sketch the proof of our main technical result that random sparse designs are erasure-robust. Finally, we discuss how this result leads to stronger lower bounds against Preconditioned Lasso.

## 4.1   Lower Bounds via Sparse Designs.

The hard covariance matrix constructed in [32] to prove Theorem 1.1 is defined as $\tilde{\Sigma} = \tilde{\Theta}^{-1}$ where $\tilde{\Theta} = \Theta + \epsilon I$ and $\Theta = M^T M$, for a rectangular matrix $M$. Note that for small $\epsilon > 0$, this covariance is very ill-conditioned, so long as $M$ has non-trivial kernel. To prove that all Preconditioned Lasso algorithms with $m$ samples fail to recover $k$-sparse signals, these three properties are used:

1. The rows of $M$ are $k$-sparse,
2. $\dim \ker M \geq 2m$,
3. $\ker M$ is bounded away from all $(n/k) \log(n)$-sparse vectors.

The first property is self-explanatory. One way to achieve the second property is if $M$ has at most $n - 2m$ rows. And the third property, in compressed sensing, is essentially what a design matrix needs to satisfy to information-theoretically enable $(n/k) \log(n)$-sparse recovery. Thus, to show that $\Omega(n)$ samples are needed to recover $\mathrm{polylog}(n)$-sparse signals, $M$ must be a sparse, compressive matrix which (as a design matrix) enables the recovery of $n/\mathrm{polylog}(n)$-sparse signals.

How do these properties imply that for every preconditioner $S$, there is a bad $k$-sparse signal? By the first property, the rows of $M$ are valid signals. For each row $M_i$, if it is not a bad signal for $S$-preconditioned Lasso, then it can be shown to induce a certain constraint on $S$: namely, that every column of $S$ either has small magnitude or is nearly orthogonal to $M_i$. So if none of the rows of $M_i$ are bad signals, then every column of $S$ either has small magnitude or lies near $\ker M$, in which case by the third property it must be $(n/k) \log(n)$-dense. Roughly speaking, this structure can be used together with the second property to show that a $k$-sparse signal with uniformly random support causes the Preconditioned Lasso to fail.

**A hard signal distribution?** The above proof shows that for any preconditioner, either it fails (with high probability) on a random $k$-sparse signal, or there *exists* some row of $M$ on which it fails. If we want a signal distribution that is uniformly hard, it's therefore natural to equiprobably pick either (a) a random row of $M$, or (b) a random $k$-sparse signal. But then the above proof only

implies that for this signal distribution, for any preconditioner, the Preconditioned Lasso fails with probability $\Omega(1/n)$. Moreover, it's not clear whether the failure probability can be improved under just the above assumptions: consider the case that for some preconditioner, just a few rows of $M$ are bad signals. Then the likelihood that one of these rows is chosen as the signal is only $O(1/n)$. Moreover, the columns of $S$ are now only forced to be orthogonal to most rows of $M$, not all. As a result, the columns may have large magnitude and yet fail to be dense, because for sparse matrices like $M$, it's possible to adversarially delete a few rows so that the kernel of the remaining rows contains sparse vectors. This is an obstacle to proving that such a preconditioner must fail on a sparse signal with uniformly random support.

To circumvent this obstacle, we need to show that a preconditioner $S$ which has columns orthogonal to most rows of $M$, but not all, still has useful structure. As we suggested earlier, this can be done by reasoning about sparse compressive matrices under adversarial deletions.

## 4.2 Erasure-robustness

We will return to the lower bound problem in the next section of the overview, but for now focus on the core technical result about partial recovery with adversarial erasures. Based on the discussion after Theorem 2.3, we need to solve the following problem.

Let $M$ be an $m \times n$ sparse random Bernoulli matrix with parameter $p = \Theta(\log n)/m$. We want to show that with high probability, $M$ supports erasure-robust partial sparse recovery: that is, for any set $B \subseteq [m]$ of "bad equations", there is a small set $C \subseteq [n]$ such that if $x \in \mathbb{R}^n$ is $\tau$-sparse, then

$$\|x_{C^c}\|_2 \leq \text{poly}(n) \cdot \|M_{B^c}x\|_\infty .$$

**Erasure-robustness: the exact case.** For simplicity, in this proof sketch we start by considering the *exact* case, where $M_{B^c}x = 0$, and we want to show that either $|\text{supp}(x)| \geq \tau$ or $\text{supp}(x) \subseteq C$. Without erasures (i.e. $B = \emptyset$), this property follows for $C = \emptyset$ by the fact that the adjacency graph of $M$ is with high probability a unique-neighbor expander.[8] Concretely, because the graph is a $(1 - \epsilon)d$ expander for a small constant $\epsilon > 0$, any set $S \subseteq [n]$ of size at most $\tau := O(m/\log(n))$ has at least $(1 - O(\epsilon))d|S|$ unique neighbors in $[m]$. Moreover, if $j \in [m]$ is a unique neighbor of $\text{supp}(x)$ for some vector $x \in \mathbb{R}^n$, then $M_j x \neq 0$. Thus, if $Mx = 0$ then $\text{supp}(x)$ must have no unique neighbors, so either $|\text{supp}(x)| \geq \tau$ or $x = 0$.

However, this argument breaks down in the presence of adversarial erasures. All that can be said is that if $M_{B^c}x = 0$ then $\text{supp}(x)$ must have no unique neighbors in $B^c$. By the unique neighbor lower bound, it does follow that either $|\text{supp}(x)| \geq \tau$ or $|\text{supp}(x)| \leq O(|B|/d)$ — this can be thought of as a kind of *density amplification* result for $\ker M_{B^c}$, since it eliminates the possibility of any vector in the kernel having an intermediate density. Unfortunately, this does not directly imply erasure-robustness, because we need a single set $C$ that contains the supports of all sparse vectors in $\ker M$, not a different $C$ for each $x$. (For example, if we allow $C = \text{supp}(x)$ then the result is not very interesting.) Moreover, it's not clear that anything useful can be said about the vertex set $\text{supp}(x)$: certainly many vertices in $\text{supp}(x)$ must be adjacent to "bad" equations, but it's conceivable that other vertices could be farther away. Pictorially, one possible case (of many) is that $B$ could be chosen as the set of "boundary" equations of a ball subgraph; then $\ker M_{B^c}$ certainly contains a vector supported on the ball, which is not actually contained in the neighborhood of $B$.

Given the above obstacles, one approach is to show that although $\text{supp}(x)$ may not be contained in the neighborhood of $B$, it must be contained in a distance-$r$ ball around $B$, for some small but super-constant $r$. The argument is that if there is a vertex of $\text{supp}(x)$ which is distance greater than $r$ from $B$, then by iteratively growing neighborhoods of the vertex until $B$ is reached, the support must have size at least $d^r$, and a contradiction is reached if $d^r > |B|/d$, because then $B$ cannot contain all unique neighbors of $\text{supp}(x)$. Unfortunately, the constructed set $C$ (the distance-$r$ ball around $B$) then has size $|B| \cdot (d^2)^r \approx |B|^3$, since the distance metric is that two coordinates are adjacent if they share an equation. This is much larger than the desired bound ($O(|B|)$) and in particular, too large to use in our ultimate lower bound application.

In summary, to get the linear bound claimed in Theorem 2.3, we need a different argument. The key idea is to exploit *linearity*. We want to show that the union $U$ of supports of all $\tau$-sparse vectors in

---

[8]We note that this initial part of the argument (the case without erasures) is quite reminiscent of arguments used in the analysis of LDPC codes (see e.g. [41]).

ker $M_{B^c}$ has small size. We've seen that for any fixed $x$, there is a *density amplification* result: if $M_{B^c}x = 0$ and $x$ is $|B|/d$-dense, then $x$ must be $\tau$-dense. So take vectors $x^{(1)}, \ldots, x^{(n)} \in \ker M_{B^c}$ which are $\tau$-sparse (and therefore $|B|/d$-sparse) and which cover $U$. Now observe that since $x^{(1)}$ and $x^{(2)}$ are $O(|B|/d)$-sparse, any linear combination $c_1 x^{(1)} + c_2 x^{(2)}$ must be $2|B|/d$-sparse. But $c_1 x^{(1)} + c_2 x^{(2)} \in \ker M_{B^c}$ by linearity. So if $2|B|/d < \tau$, then by the (contrapositive of the) density amplification result, we in fact know that the sum is $|B|/d$ sparse! Inductively, it follows that any linear combination $c_1 x^{(1)} + \cdots + c_n x^{(n)}$ is $|B|/d$-sparse. But for generic $c_1, \ldots, c_n$, we have

$$\operatorname{supp}(c_1 x^{(1)} + \cdots + c_n x^{(n)}) = \bigcup_{i=1}^{n} \operatorname{supp}(x^{(i)}) = U.$$

This shows that in fact we can find a set $C$ of size $O(|B|/d)$ satisfying the desired property.

**Erasure robustness: the general case.** Note that the above argument was when $M_{B^c}x = 0$. The proof for the general case, when $M_{B^c}x$ is small but not nonzero, uses the same insight with several complications. First, we need a quantitative density amplification lemma which states that if $M_{B^c}x$ is small and $x$ has more than $|B|$ coordinates with magnitude exceeding some threshold $\delta$, then we can trade off density for magnitude, i.e. find $\tau$ coordinates with magnitude exceeding $\delta/\operatorname{poly}(n)$. To prove this without losing a superpolynomial factor on the threshold, we actually need the graph to satisfy a stronger property than just expansion: we also need that for any two disjoint sparse sets $S, T \subseteq [n]$, the intersection of their neighborhoods has size only $O(\sqrt{d} \cdot \max(|S|, |T|))$. Note that expansion would only give a bound of $O(\epsilon d \max(|S|, |T|))$. Nonetheless, it can be proven that the random sparse adjacency matrix of $M$ satisfies the desired stronger property with high probability.

Second, the iterative addition procedure in the noiseless case requires a modification for the noisy case; each addition causes the quantitative threshold to decay, and after $n$ additions it would decay by a factor superpolynomial in $n$. Instead, we add the vectors $x^{(1)}, \ldots, x^{(n)}$ recursively according to a $d$-ary tree. This tree has depth only $\log_d n$, which allows the decay to be controlled to only a $\operatorname{poly}(n)$ factor, proving Theorem 2.3.

### 4.3 Stronger lower bound via erasure-robustness

We now return to the problem of proving hardness against Preconditioned Lasso. Theorem 2.3 can be used to show that for an appropriately chosen $M$, if the number of rows of $M$ that are bad signals for $S$-preconditioned Lasso is at most $n/\operatorname{polylog}(n)$, then there is a set $C \subseteq [n]$ of size $n/\operatorname{polylog}(n)$ such that each column of $S$ is either $n/\operatorname{polylog}(n)$-dense, or has small magnitude on coordinates outside the set $C$. This is precisely the structure lemma we need for preconditioners that succeed on most rows of $M$: it crucially allows for a nearly-linear number of rows of $M$ that are bad signals, although in exchange there is a set $C$ of sublinear size where we cannot control the columns of $S$ (the corresponding structure lemma in [32] could not tolerate any bad rows). We also show that in this situation, the number of dense columns of $S$ must be $\Omega(n)$.

With these results, we can prove our lower bounds. First, to prove our lower bound against invertible preconditioners (Theorem 2.1), we define a distribution over $\operatorname{polylog}(n)$-sparse signals by taking $w^*$ to be the sum of $\operatorname{polylog}(n)$ random rows of $M$, plus an infinitesimal uniformly random $\operatorname{polylog}(n)$-sparse vector. Under certain conditions, if at least one of the rows in the sum is a bad signal, then the sum must also be a bad signal. With this amplification (at the cost of a $\operatorname{polylog}(n)$ factor in sparsity), any invertible preconditioner must fail with probability $1 - o(1)$: either there are $\Omega(n/\operatorname{polylog}(n))$ bad rows of $M$, in which case the sum of the chosen rows is a bad signal with high probability, and the infinitesimal perturbation does not affect the program failure. Or, $S$ has many dense columns, in which case $S^T w^*$ is dense due to the perturbation. By a dimension-counting argument (which crucially uses invertibility of $S$), this implies that there exists a feasible direction of improvement for the program objective.

Extending the lower bound to rectangular preconditioners is more involved and involves generalizations of techniques from [32]. The factor of $1/2$ in Theorem 2.2 arises because we are not able to construct a single signal distribution that causes failure of both "incompatible" preconditioners (i.e. for which more than $n/\operatorname{polylog}(n)$ rows of $M$ are bad signals) and "compatible" preconditioners (for which at most $n/\operatorname{polylog}(n)$ rows of $M$ are bad signals, so the structure lemma applies) with high probability. Instead, we take a mixture of the two cases' hard distributions: either a sum of rows

of $M$, or a uniformly random sparse vector. Due to the lack of invertibility of the preconditioners, the second case is no longer a simple dimension-counting argument. In [32], the proof crucially relies on a "projection lemma" which states that if $\dim \ker M \geq 2m$, then any fixed direction is unlikely to align with the span of the covariates. Since our structure lemma has no control over the preconditioner columns in the subspace indexed by the set $C$, we prove a generalized projection lemma which states that alignment is unlikely even on $C^c$. This yields Theorem 2.2.

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
