# A   Organization

In Appendix B, we discuss notation and collect important definitions, e.g. of erasure-robustness. In Appendix C, we prove that random sparse compressive matrices are erasure-robust (Theorem 2.3). In Appendix D, we prove the key structure lemma that connects erasure-robustness with lower bounds against Preconditioned Lasso. In Appendix E, we then use this lemma together with our result about erasure-robustness to construct example distributions that are provably hard against Preconditioned Lasso algorithms (Theorem 2.1 and Theorem 2.2).

# B   Preliminaries

For vectors $x, y$ we denote the inner product as $\langle x, y \rangle = x^T y$. For a matrix $M$, we let $\mathrm{span}(M)$ and $\ker(M)$ denote the row space and null space of $M$ respectively. For a matrix $M \in \mathbb{R}^{n \times p}$, a vector $v \in \mathbb{R}^n$, and a subset $U \subseteq [n]$, we say that $M_U$ is the $|U| \times p$ matrix consisting of the rows of $M$ indexed by $U$; similarly, $v_U$ consists of the entries of $v$ indexed by $U$; and the complement of $U$ in $[n]$ is $U^c = \bar{U} = [n] \setminus U$. We use the standard notation for vector norms that $\|v\|_p = \left(\sum_{i=1}^n |v_i|^p\right)^{1/p}$.

## B.1   Preconditioners and the Preconditioned Lasso

**Definition 2.** For $n, s \in \mathbb{N}$, a *preconditioner* is a matrix $S \in \mathbb{R}^{n \times s}$ with $\ker(S^T) = \{0\}$. The *$S$-preconditioned-Lasso* on samples $(X, y)$ is the convex program

$$\hat{w} \in \operatorname*{argmin}_{w \in \mathbb{R}^n : Xw = y} \left\| S^T w \right\|_1 .$$

In our lower bounds, we will say that the $S$-preconditioned Lasso *fails* if the true signal vector $w^*$ is not contained in the set of optimal solutions to the program (i.e. some other vector achieves strictly smaller objective value). As a result, some restriction on $\ker(S^T)$ is necessary, to rule out programs with multiple (and therefore infinitely many) co-optimal solutions. In [32], it is only assumed that $S$ is not identically zero; this suffices for their purposes because they are allowed to pick the true signal depending on $S$. However, we want an algorithm-independent distribution over signals. If $S^T$ is nonzero in only a few directions, then it's obviously impossible to cause the $S$-preconditioned Lasso to fail (in our strong sense) with non-trivial probability without knowing $S$. Nonetheless such a program is clearly not useful.

More to the point, any matrix $S$ where $\ker(S^T)$ is non-trivial can be perturbed infinitesimally (possibly by adding columns) so that $\ker(S^T)$ is trivial, but so that if the original program *uniquely* recovered the true signal, then the new program still does so.

## B.2   Supports, erasure-robustness, and quantitative density

**Definition 3.** For $x \in \mathbb{R}^n$ and $\delta > 0$, define the $\delta$-support of $x$ to be

$$\mathrm{supp}_\delta(x) := \{i \in [n] : |x_i| \geq \delta\}.$$

We sometimes refer to $\mathrm{supp}_\delta(x)$ as a *quantitative support* of $x$ with threshold $\delta$. The support of $x$ is $\mathrm{supp}(x) := \|x\|_0 = \{i \in [n] : |x_i| > 0\}$.

**Definition 4.** Let $M \in \mathbb{R}^{m \times n}$ be a matrix. We say that $M$ is $(b, b', \eta, \tau)$-*erasure-robust* if for any set $B \subseteq [m]$ of size $|B| \leq b$, there is a set $C \subseteq [n]$ of size $|C| \leq b'$ with the following property: for every $x \in \mathbb{R}^n$, either:

- $|\mathrm{supp}(x)| = \|x\|_0 \geq \tau$, or

- $\|x_{C^c}\|_2 \leq \eta \|M_{B^c} x\|_\infty$.

If $M$ satisfies this property, we say that it tolerates $b$ erasures and sparsity level $\tau$, with only $b'$ unidentifiable coordinates.

**Definition 5.** For any subspace $V \subseteq \mathbb{R}^n$ and vector $x \in \mathbb{R}^n$, the (Euclidean) distance from $x$ to $V$ is

$$\mathrm{dist}(x, V) := \inf_{v \in V} \|x - v\|_2 = \left\|\mathrm{Proj}_{V^\perp} x\right\|_2 .$$

**Definition 6.** Let $V \subseteq \mathbb{R}^n$ be a subspace. We say that $V$ is $(\delta, \eta, \tau)$-*quantitatively dense* if for any set $C \subseteq [n]$ of size $|C| \leq \tau$, for any $x \in \mathbb{R}^n$ with $\mathrm{dist}(x, V) \leq \delta \|x\|_2$, it holds that $\|x_{C^c}\|_2 \geq \eta \|x\|_2$.

### B.3   Random Matrix Theory

We will use the following standard bound on the singular values of Gaussian random matrices.

**Theorem B.1** (e.g. Corollary 5.35 in [50])**.** *Let $n, N \in \mathbb{N}$. Let $A \in \mathbb{R}^{N \times n}$ be a random matrix with entries i.i.d. $N(0, 1)$. Then for any $t > 0$, it holds with probability at least $1 - 2\exp(-t^2/2)$ that*

$$\sqrt{N} - \sqrt{n} - t \leq \sigma_{min}(A) \leq \sigma_{max}(A) \leq \sqrt{N} + \sqrt{n} + t.$$

**Theorem B.2** (See e.g. Exercise 4.7.3 of [51])**.** *Suppose $X_1, \ldots, X_m \sim N(0, \Sigma)$ with $\Sigma : n \times n$ a positive definite matrix, $t > 0$ and $m = \Omega(n + t^2)$. Let $\hat{\Sigma} = \frac{1}{m} \sum_i X_i X_i^T$. Then with probability at least $1 - 2\exp(-t^2/2)$,*

$$(1 - \epsilon)\Sigma \preceq \hat{\Sigma} \preceq (1 + \epsilon)\Sigma$$

*with $\epsilon = O(\sqrt{n/m} + \sqrt{t^2/m})$.*

## C   Erasure-Robustness via Expanders

In this section, we construct a sparse and compressive measurement matrix $M$ which satisfies erasure-robustness with near-linear erasure tolerance and sparsity level, and so that the number of unidentifiable coordinates is only a constant multiple of the number of erasures. In fact, we'll show that a sparse random Bernoulli matrix has the desired property with high probability, proving Theorem 2.3 and also enabling our construction of a hard distribution family for Preconditioned Lasso.

Recall that the erasure-robustness property is defined as follows: if $B$ is an adversarially chosen subset of indices of rows of $M$, then $M_{B^c}$ is still a good measurement matrix, meaning that any sparse signal $x$ can still be approximately recovered from noisy measurements $M_{B^c}x + \xi$, except for the entries of $x$ in a small set of input coordinates $C = C(B)$ that is *independent of $x$*. In our construction, we will prove that $|C(B)| = O(|B|)$, which is nearly optimal because $M$ is sparse, and erasing all equations involving one particular coordinate of the signal clearly makes it impossible to recover that coordinate.

**Notation.**   Let $n, m \in \mathbb{N}$. Let $M \in \{0, 1\}^{m \times n}$ be a binary matrix. The matrix $M$ defines a bipartite graph between a set of "equations" $[m]$ and a set of "coordinates" $[n]$. For $S \subseteq [n]$ and $E \subseteq [m]$, we can define neighborhoods

$$N(S) = \{i \in [m] : \exists j \in S : M_{ij} = 1\}$$

and

$$N'(E) = \{j \in [n] : \exists i \in E : M_{ij} = 1\}.$$

For $S \subseteq [n]$, further define the "unique neighborhood" of $S$ to be

$$U(S) = \{i \in [m] : \|M_{iS}\|_0 = 1\}.$$

### C.1   Deterministic Conditions for Erasure-Robustness

We start by proving that erasure-robustness holds whenever $M$ satisfies certain deterministic conditions. Throughout this section, we make the following assumption encapsulating the needed properties: approximate regularity, vertex expansion, and a bounded intersection property.

**Assumption 1.**   For some $d, k \in \mathbb{N}$ and $\epsilon > 0$, suppose that $M$ satisfies the following deterministic conditions:

- **(Degree bounds)** For all $i \in [m]$ and $j \in [n]$,

$$|N(j)| \leq (1 + \epsilon)d$$

  and

$$|N'(i)| \leq (1 + \epsilon)(n/m)d.$$

- **(Expansion)** For all $S \subseteq [n]$ with $|S| \leq k$,

$$|N(S)| \geq (1 - \epsilon)d|S|.$$

- **(Bounded intersection)** For all disjoint $S, T \subseteq [n]$ with $|S|, |T| \leq k$,

$$|N(S) \cap N(T)| \leq \frac{\sqrt{d}}{8} \max(|S|, |T|).$$

Note that the first two conditions imply a weaker form of the bounded intersection property, with $2\epsilon d$ instead of $\sqrt{d}/8$. However, we need the stronger bound to prove our result.

An immediate corollary of the first two conditions is a unique-neighbor lower bound.

**Corollary C.1** (Folklore). *For all $S \subseteq [n]$ with $|S| \leq k$,*

$$|U(S)| \geq (1 - 3\epsilon)d|S|.$$

*Proof.* By the degree bound assumption, we have $\sum_{j \in S} |N(j)| \leq (1 + \epsilon)d|S|$. Together with the expansion assumption, it follows that $\sum_{j \in S} |N(j)| - |N(S)| \leq 2\epsilon d|S|$. Every unique neighbor counts once in both terms, and every non-unique neighbor counts at least twice in the first term but only once in the second term. So there are at most $2\epsilon d|S|$ non-unique neighbors, which means that $|U(S)| \geq |N(S)| - 2\epsilon d|S| \geq (1 - 3\epsilon)d|S|$. $\square$

**Density amplification.** For exact solutions $x \in \mathbb{R}^n$ to $M_{B^c} x = 0$, it's easy to see that the unique neighbor lower bound implies a density amplification statement: if $|\operatorname{supp}(x)| > |B|/((1 - 3\epsilon)d)$ and $|\operatorname{supp}(x)| \leq k$, then by Corollary C.1, $U(\operatorname{supp}(x)) \cap B^c$ must be nonempty, which contradicts $M_{B^c} x = 0$. Thus, $|\operatorname{supp}(x)| > |B|/((1 - 3\epsilon)d)$ implies $|\operatorname{supp}(x)| > k$.

For approximate solutions, i.e. when $\|M_{B^c} x\|_\infty \leq \delta$, this argument does not quite work: if some equation $i \in B^c$ has a unique neighbor in $\operatorname{supp}(x)$, this is not necessarily a contradiction because the value of $x$ at that coordinate may be within the error tolerance $\delta$. Thus, we can only hope for amplification if we have a lower bound on $|\operatorname{supp}_\gamma(x)|$, the number of coordinates of $x$ exceeding some threshold $\gamma \gg \delta$ in magnitude. However, there is now another problem: even if some equation has a unique neighbor in $\operatorname{supp}_\gamma(x)$, it may have many neighbors where $x$ has magnitude just slightly less than $\gamma$, so that these coordinates together cancel out the large coordinate. If this happens, then that equation must neighbor both $\operatorname{supp}_\gamma(x)$ and $\operatorname{supp}_{\gamma/(2d)}(x) \setminus \operatorname{supp}_\gamma(x)$. Naively, we can use expansion to bound the number of such equations by $2\epsilon d \cdot |\operatorname{supp}_{\gamma/(2d)}(x)|$. However, combined with the unique neighbor lower bound, this only shows that $|\operatorname{supp}_{\gamma/(2d)}(x)| \geq \Omega(1/\epsilon) \cdot |\operatorname{supp}_\gamma(x)| - |B|$. That is, density is amplified by a constant factor, but the threshold decays by a factor of $2d = \Omega(\log n)$.

This tradeoff is not good enough for our purposes (tolerating inverse-polynomial measurement error). The following lemma uses the bounded intersection property to prove a better tradeoff:

**Lemma C.2** (One-step quantitative density amplification). *Suppose that $\epsilon \leq 1/12$ and $d \geq 2$. Let $x \in \mathbb{R}^n$ and $\delta > 0$. If $\|M_{B^c} x\|_\infty \leq \delta$, then*

$$|\operatorname{supp}_\delta x| \geq \min\left(6\sqrt{d}|\operatorname{supp}_{2d\delta} x| - |B|, k\right).$$

*Proof.* Let $S = \operatorname{supp}_{2d\delta} x$ and let $T = \operatorname{supp}_\delta x \setminus S$. If $|\operatorname{supp}_\delta x| > k$ then the claim holds, so suppose that $|\operatorname{supp}_\delta x| \leq k$. On the one hand, by Assumption 1, we have that

$$|N(S) \cap N(T)| \leq \frac{\sqrt{d}}{8} \max(|S|, |T|) \leq \frac{\sqrt{d}}{8} |\operatorname{supp}_\delta x|.$$

On the other hand, we claim that $U(S) \setminus B \subseteq N(S) \cap N(T)$. Clearly $U(S) \setminus B \subseteq N(S)$, so it remains to show inclusion in $N(T)$. Suppose that this is false. Then pick some $i \in U(S) \setminus (B \cup N(T))$.

Since $i \in U(S)$, there is a unique $j \in S$ such that $M_{ij} = 1$. Now

$$|M_i x| \geq |x_j| - \sum_{j' \neq j} |M_{ij'} x_{j'}|$$
$$\geq 2d\delta - \sum_{j' \neq j: M_{ij'} = 1} \delta$$
$$\geq (1 - \epsilon)d\delta.$$

The first inequality is by the triangle inequality. The second inequality uses $j \in S$ and the definition of $S$; it also uses that any $j' \neq j$ with $M_{ij'} = 1$ satisfies $j' \notin T$ (since $i \notin N(T)$) and $j' \notin S$ (since $j$ is the unique neighbor of $i$ in $S$): thus $j' \notin \operatorname{supp}_\delta x$, so $|x_{j'}| \leq \delta$. The last inequality is by the degree bound on $i$.

But since $\epsilon < 1 - 1/d$, the resulting inequality $|M_i x| \geq (1-\epsilon)d\delta$ contradicts the lemma assumption that $\|M_{B^c} x\|_\infty \leq \delta$, since $i \notin B$. So in fact the initial claim that

$$U(S) \setminus B \subseteq N(S) \cap N(T)$$

was true. As a result,

$$|\operatorname{supp}_\delta x| \geq \frac{8|N(S) \cap N(T)|}{\sqrt{d}} \geq \frac{8|U(S) \setminus B|}{\sqrt{d}} \geq 6\sqrt{d}|S| - |B|$$

where the final inequality uses Corollary C.1 and the assumption that $\epsilon \leq 1/12$. $\qquad\square$

Recursively applying Lemma C.2 gives our quantitative analogue of the noiseless density amplification: if $\|M_{B^c} x\|_\infty$ is small (i.e. $x$ approximately satisfies the "good" equations of $M$), then if $x$ has at least $|B|$ not-too-small entries, it must in fact have at least $k$ not-too-small entries. Moreover, the threshold defining "not-too-small" only decays by a polynomial factor.

**Corollary C.3** (Quantitative density amplification). *Suppose that $\epsilon \leq 1/12$ and $d \geq 16$. Let $f(n, d) = 2dn^2$. Let $x \in \mathbb{R}^n$ and $\delta > 0$. If $\|M_{B^c} x\|_\infty \leq \delta$, and $|\operatorname{supp}_{f(n,d)\delta}(x)| \geq |B|$, then $|\operatorname{supp}_\delta(x)| \geq k$.*

*Proof.* For any $\delta' \geq \delta$ with $|\operatorname{supp}_{2d\delta'}(x)| \geq |B|$, Lemma C.2 implies that

$$|\operatorname{supp}_{\delta'}(x)| \geq \min\left(\sqrt{2d}|\operatorname{supp}_{2d\delta'}(x)|, k\right).$$

Let $t \in \mathbb{N}$ be maximal subject to $|\operatorname{supp}_{(2d)^t \delta}(x)| \geq |B|$. Applying the inequality to all $\delta' \in \{\delta, 2d\delta, \ldots, (2d)^{t-1}\delta\}$ gives that

$$|\operatorname{supp}_\delta(x)| \geq \min((\sqrt{2d})^t, k).$$

But since $|\operatorname{supp}_{f(n,d)\delta}(x)| \geq |B|$, we know that $(2d)^{t+1} > f(n, d) = 2dn^2$, so $(\sqrt{2d})^t \geq n$. Thus, $|\operatorname{supp}_\delta(x)| \geq k$. $\qquad\square$

**Bounding the union of supports.** We've shown that if a single vector $x$ has quantitative support larger than $|B|$ at some threshold, then we can amplify the density by decreasing the threshold (so long as $x$ is an "approximate solution", i.e. $\|M_{B^c} x\|_\infty$ is sufficiently small). Ultimately, we want to bound the union of the quantitative supports over all $k$-sparse vectors for which $M_{B^c} x$ is small. The key step is the following lemma, which shows that adding quantitatively sparse approximate solutions preserves quantitative sparsity (although we have to decrease the threshold by a factor polynomial in the number of terms in the sum).

The lemma is proved recursively: we divide the terms into $O(\sqrt{d})$ groups, apply the inductive hypothesis to bound the supports of the group sums, and then add up the $O(\sqrt{d})$ sums. When there are only $O(\sqrt{d})$ terms, the proof is via Lemma C.2: if the sum has quantitative support larger than $|B|$, then by amplification it must have quantitative support at least $O(\sqrt{d}|B|)$. But by the triangle inequality, adding up $r$ terms can only increase the quantitative support size by a factor of $r$. For an appropriate choice of constants, this yields a contradiction, proving the desired claim.

**Lemma C.4.** *Suppose that $\epsilon \leq 1/12$ and $k \geq 4\sqrt{d}|B|$ and $d \geq 16$. Let $\delta > 0$ be a threshold. Let $r \leq (\sqrt{2d})^t$ for some $t \in \mathbb{N}$. Let $x_1, \ldots, x_r \in \mathbb{R}^n$ satisfy $\|M_{B^c}x_i\|_\infty \leq \delta/\sqrt{2d}$ and $|\operatorname{supp}_\delta x_i| \leq |B|$ for all $i \in [r]$. Then*

$$\left| \operatorname{supp}_{(2d)^{3t/2}\delta} \sum_{i=1}^r x_i \right| \leq |B|.$$

*Proof.* We induct on $t$. If $t = 0$ then $r \leq 1$, and the claim is trivial. Suppose $t = 1$. For the sake of contradiction, assume that

$$\left| \operatorname{supp}_{(2d)^{3/2}\delta} \sum_{i=1}^r x_i \right| > |B|.$$

We know that

$$\left\| M_{B^c} \sum_{i=1}^r x_i \right\|_\infty \leq \delta r/\sqrt{2d} \leq \delta.$$

So by Lemma C.2,

$$\left| \operatorname{supp}_{\sqrt{2d}\delta} \sum_{i=1}^r x_i \right| \geq \min(6\sqrt{d}|B| - |B|, k) \geq 4\sqrt{d}|B|.$$

But

$$\operatorname{supp}_{\sqrt{2d}\delta} \sum_{i=1}^r x_i \subseteq \bigcup_{i=1}^r \operatorname{supp}_\delta x_i$$

by the triangle inequality and since $r \leq \sqrt{2d}$: if some coordinate $j \in [n]$ is contained in none of the sets $\operatorname{supp}_\delta x_i$, then $|x_{ij}| < \delta$ for all $i \in [r]$, so $|\sum_{i=1}^r x_{ij}| < r\delta \leq \sqrt{2d}\delta$. Now, the right-hand-side set in the above inclusion has size at most $|B|r \leq \sqrt{2d}|B|$, which contradicts the previous bound we obtained on the size of the left-hand-side set. This proves the lemma statement for $t = 1$.

Now suppose $t \geq 2$. Suppose that the lemma holds for all $r \leq (\sqrt{2d})^{t-1}$. Pick some $x_1, \ldots, x_r \in \mathbb{R}^n$ with $r \leq (\sqrt{2d})^t$, satisfying $\|M_{B^c}x_i\|_\infty \leq \delta/\sqrt{2d}$ and $|\operatorname{supp}_\delta x_i| \leq |B|$ for all $i \in [r]$. Define vectors $y_1, \ldots, y_{\sqrt{2d}} \in \mathbb{R}^n$ by $y_b = \sum_{i=1+(b-1)(\sqrt{2d})^{t-1}}^{\min(b(\sqrt{2d})^{t-1}, r)} x_i$. By the inductive hypothesis applied to each of these smaller sums, we have for all $b \in [\sqrt{2d}]$ that

$$|\operatorname{supp}_{(2d)^{3(t-1)/2}\delta} y_b| \leq |B|.$$

Moreover for each $b \in [\sqrt{2d}]$,

$$\|M_{B^c}y_b\|_\infty \leq \sum_{i=1+(b-1)(\sqrt{2d})^{t-1}}^{\min(b(\sqrt{2d})^{t-1}, r)} \|M_{B^c}x_i\|_\infty \leq (\sqrt{2d})^{t-1}\delta/\sqrt{2d} \leq (2d)^{3(t-1)/2}\delta/\sqrt{2d}.$$

By the inductive hypothesis applied to $y_1, \ldots, y_{\sqrt{2d}}$ with threshold $(2d)^{3(t-1)/2}\delta$, we have that

$$\left| \operatorname{supp}_{(2d)^{3t/2}\delta} \sum_{b=1}^{\sqrt{2d}} y_b \right| \leq |B|$$

as desired. $\qquad\square$

The previous lemma lets us bound the quantitative support of a sum of vectors. To show that this bounds the union of the supports, we also need the following lemma, which is proved by the probabilistic method.

**Lemma C.5.** *Let $\delta > 0$. Let $x_1, \ldots, x_r \in \mathbb{R}^n$ and let $V = \bigcup_{i=1}^r \operatorname{supp}_\delta x_i$. Then there is some $\sigma \in \{-1, 1\}^r$ such that*

$$\left| \operatorname{supp}_\delta \sum_{i=1}^n \sigma_i x_i \right| \geq \frac{|V|}{2}.$$

*Proof.* Suppose we choose $\sigma$ uniformly at random. For any $j \in V$, let $a \in [r]$ be such that $|x_{aj}| \geq \delta$. For any $\sigma$, if $|\sum \sigma_i x_{ij}| < \delta$, then if we flip $\sigma_a$ to obtain a sign vector $\sigma'$, it's necessary that $|\sum \sigma_i' x_{ij}| \geq \delta$. Thus, conditioned on $\sigma_a$ it holds that $|\sum_i \sigma_i x_{ij}| \geq \delta$ with probability at least $1/2$, so unconditionally it holds with probability at least $1/2$. Thus,

$$\mathbb{E}_{\sigma \sim \{-1,1\}^r} \left| \operatorname{supp}_\delta \sum_{i=1}^r \sigma_i x_i \right| \geq \frac{|V|}{2}.$$

So there must exist at least one $\sigma \in \{-1,1\}^r$ satisfying the desired inequality. $\qquad \square$

We can now bound the union of the quantitative supports of sparse approximate solutions to $M_{B^c} x = 0$, by picking $n$ vectors whose quantitative supports cover the union, and adding them up with the signs suggested by Lemma C.5.

**Lemma C.6.** *Suppose that $\epsilon \leq 1/12$ and $|B| \leq k/(4\sqrt{d})$ and $d \geq 16$. Let $\delta > 0$. Define*

$$S = \{x \in \mathbb{R}^n : \|M_{B^c} x\|_\infty \leq \delta/\sqrt{2d} \wedge |\operatorname{supp}_\delta(x)| \leq |B|\}$$

*and define*

$$C = \bigcup_{x \in S} \operatorname{supp}_{g(n,d)\delta}(x)$$

*where $g(n,d) = (2nd)^3$. Then $|C| \leq 2|B|$.*

*Proof.* Since $|C| \leq n$, we can find $x_1, \ldots, x_n \in \mathbb{R}^n$ such that $C = \bigcup_{i=1}^n \operatorname{supp}_{g(n,d)\delta}(x_i)$, and $\|M_{B^c} x_i\|_\infty \leq \delta/\sqrt{2d}$ and $|\operatorname{supp}_\delta(x)| \leq |B|$ for all $i \in [n]$. By Lemma C.5, there are some $\sigma_1, \ldots, \sigma_n \in \{-1, 1\}$ such that

$$\left| \operatorname{supp}_{g(n,d)\delta} \sum_{i=1}^n \sigma_i x_i \right| \geq \frac{|C|}{2}.$$

Let $t$ be minimal so that $n \leq (\sqrt{2d})^t$, and note that $(\sqrt{2d})^t \leq 2nd$, so $(2d)^{3t/2} \leq (2nd)^3 = g(n,d)$. By Lemma C.4, which is applicable since $\|M_{B^c} \sigma_i x_i\|_\infty \leq \delta/\sqrt{2d}$ and $|\operatorname{supp}_\delta \sigma_i x_i| \leq |B|$ for all $i \in [n]$, we have that

$$\left| \operatorname{supp}_{g(n,d)\delta} \sum_{i=1}^n \sigma_i x_i \right| \leq \left| \operatorname{supp}_{(2d)^{3t/2}\delta} \sum_{i=1}^n \sigma_i x_i \right| \leq |B|.$$

It follows that $|C| \leq 2|B|$. $\qquad \square$

From the above lemma, erasure-robustness of $M$ is essentially immediate; it only remains to observe that although we took the union of all approximate solutions with quantitative supports of size at most $|B|$, this is equivalent to taking the union of all approximate solutions with quantitative supports of size less than $k$ (for a slightly different threshold) due to density amplification.

**Corollary C.7** (Erasure-robustness under Assumption 1). *Suppose that $\epsilon \leq 1/12$ and $n, d \geq D_0$, for some universal constant $D_0$. For any $b \leq k/d$, the matrix $M$ is $(b, 2b, d^4 n^6, k)$-erasure-robust (Definition 4). To restate: let $B \subseteq [m]$ satisfy $|B| \leq k/d$. Then there is a set $C \subseteq [n]$ with $|C| \leq 2|B|$, such that for any $x \in \mathbb{R}^n$, either:*

- $\|x_{C^c}\|_2 \leq d^4 n^6 \|M_{B^c} x\|_\infty$, *or*
- $|\operatorname{supp}(x)| \geq k$.

*Proof.* Pick arbitrary $\delta > 0$ and define $S$ and $C$ as in Lemma C.6, which guarantees that $|C| \leq 2|B|$. For any $x \in \mathbb{R}^n$, pick some $c \in \mathbb{R}^+$ such that $\|M_{B^c} cx\|_\infty \leq \delta/f(n,d)$, where $f(n,d) = 2dn^2$. If $|\operatorname{supp}_\delta(cx)| > |B|$, then by Corollary C.3, $|\operatorname{supp}_{\delta/f(n,d)}(cx)| \geq k$. As a result, $|\operatorname{supp}(x)| = |\operatorname{supp}(cx)| \geq k$. Otherwise, we have $|\operatorname{supp}_\delta(cx)| \leq |B|$. But we also know that $\|M_{B^c} cx\|_\infty \leq \delta/f(n,d) \leq \delta/d$. Thus, $cx \in S$. As a result, by definition of $C$, we get $\operatorname{supp}_{g(n,d)\delta}(cx) \subseteq C$, where $g(n,d) = (2nd)^3$. Therefore $\|cx_{C^c}\|_\infty \leq g(n,d)\delta$, so $\|cx_{C^c}\|_2 \leq \sqrt{n} g(n,d)\delta$.

If $\|M_{B^c}x\|_\infty > 0$, then we can choose $c = \delta/(f(n,d)\|M_{B^c}x\|_\infty)$, in which case $\|x_{C^c}\|_2 \leq \sqrt{n}f(n,d)g(n,d)\|M_{B^c}x\|_\infty \leq d^4 n^6$ for sufficiently large $n,d$. And if $\|M_{B^c}x\|_\infty = 0$, then we can choose $c$ arbitrarily large, so $\|x_{C^c}\|_2 = 0$. In either case, it holds that $\|x_{C^c}\|_2 \leq d^4 n^6 \|M_{B^c}x\|_\infty$. $\qquad\square$

All that remains to prove Theorem 2.3 is showing that a random sparse binary matrix satisfies Assumption 1 with high probability. However, for the application to lower bounds against Preconditioned Lasso, we also must recall the following result (essentially due to [6] and slightly generalized in [32]), which states that under the degree bound and expansion assumptions, any vector in the kernel of $M$ must be quantitatively dense.

**Lemma C.8** (Lemma 8.7 in [32]). *Let $x \in \mathbb{R}^n$ be such that $Mx = 0$. Let $S \subseteq [n]$ with $|S| \leq k$. If $\epsilon \leq 1/17$, then*

$$\|x_S\|_1 \leq \frac{\|x\|_1}{3}.$$

## C.2 Random unbalanced bipartite graph

In this section we show that a random sparse binary matrix satisfies Assumption 1 with high probability. Let $n,d,m \in \mathbb{N}$ with $d \leq m \leq n$, and let $p = d/m$. Define a random matrix $M \in \mathbb{R}^{m \times n}$ with independent entries $M_{ij} \sim \text{Ber}(p)$.

The following result is folklore (see e.g. [1]):

**Lemma C.9** (Expansion of a random bipartite graph). *Let $\epsilon \in (0,1)$. Suppose that $p \geq 32\epsilon^{-2}(\log n)/m$. It holds with probability at least $1 - 2/n$ that for all $S \subseteq [n]$ with $|S| \leq \epsilon/(2p)$,*

$$|N(S)| \geq d(1-\epsilon)|S|.$$

*Proof.* For $1 \leq l \leq \epsilon/(2p)$ let $q_l$ be the probability that there exists some $S \subseteq [n]$ with $|S| = l$ and $|N(S)| < d(1-\epsilon)|S|$. To bound this probability, fix $S \subseteq [n]$ with $|S| = l$. For any $y \in [m]$, we have

$$\Pr[y \in N(S)] = 1 - (1-p)^l \geq 1 - e^{-pl} \geq pl - (pl)^2 \geq (1 - \epsilon/2)pl$$

so long as $0 \leq pl \leq \epsilon/2$. Thus, by the Chernoff bound,

$$\Pr[|N(S)| < (1-\epsilon)plm] \leq \exp(-(\epsilon/2)^2(1-\epsilon/2)plm/2) \leq \exp(-\epsilon^2 plm/16).$$

By the union bound over sets of size $l$, if $\epsilon^2 pm/16 \geq 2\log n$, we have that

$$q_l \leq \binom{n}{l} \exp(-\epsilon^2 plm/16) \leq \exp(l \log n - \epsilon^2 plm/16) \leq \exp(-l\log n).$$

Finally, by a union bound over $1 \leq l \leq \epsilon/(2p)$, the lemma holds with probability at least

$$1 - \sum_{l=1}^{\epsilon/(2p)} q_l \geq 1 - \sum_{l=1}^{\epsilon/(2p)} n^{-l} \geq 1 - \frac{2}{n}$$

as claimed. $\qquad\square$

We'll also need the following simple result:

**Lemma C.10** (Degree bounds). *Let $\epsilon \in (0,1)$. Suppose that $p \geq 6\epsilon^{-2}(\log n)/m$. It holds with probability at least $1 - 1/n$ that*

$$|N(x)| \leq d(1+\epsilon)$$

*for all $x \in [n]$. Similarly, it holds with probability at least $1 - 1/n$ that $|N'(y)| \leq (n/m)d(1+\epsilon)$ for all $y \in [m]$.*

*Proof.* Fix $x \in [n]$. By the Chernoff bound,

$$\Pr[|N(x)| > (1+\epsilon)pm] \leq \exp(-\epsilon^2 pm/3).$$

Since $\epsilon^2 pm/3 \geq 2\log n$, this bound is at most $1/n^2$. Union bounding over $x \in [n]$ completes the proof of the first claim.

Similarly, fix $y \in [m]$. By the Chernoff bound,

$$\Pr[|\{x : y \in N(x)\}| > (1 + \epsilon)pn] \leq \exp(-\epsilon^2 pn/3).$$

Since $\epsilon^2 pn/3 \geq 2\log n$, this bound is at most $1/n^2$, and union bounding over $y \in [m]$ completes the proof. $\qquad\square$

To prove the last condition of Assumption 1, we need the following version of the Chernoff-Hoeffding bound:

**Lemma C.11** (Chernoff-Hoeffding). *Let $X_1, \ldots, X_n$ be i.i.d. random variables with values in $\{0,1\}$. Let $\mu = \mathbb{E}\sum X_i$. Then for any $t > 0$,*

$$\Pr\left[\sum_{i=1}^{n} X_i > t\right] \leq \exp(-t\log(t/(\mu e))).$$

Now, the proof of the bounded intersection property is analogous to the proof of expansion.

**Lemma C.12.** *Suppose that $d \geq \log^2 n$ and $n$ is sufficiently large. It holds with probability at least $1 - 2/m^2$ that for all disjoint $S, T \subseteq [n]$ with $|S|, |T| \leq m/d^3$,*

$$|N(S) \cap N(T)| \leq \frac{\sqrt{d}}{8}\max(|S|, |T|).$$

*Proof.* Fix $1 \leq l \leq m^3/(n^2 d^7)$. Let $q_l$ be the probability that some sets $S, T$ of size exactly $l$ violate the inequality. Fix disjoint $S, T \subseteq [n]$ with $|S|, |T| = l$. For any $i \in [m]$, we have that

$$\Pr[i \in N(S)] = 1 - (1-p)^l \leq pl.$$

Thus, since $S$ and $T$ are disjoint,

$$\Pr[i \in N(S) \cap N(T)] \leq p^2 l^2.$$

Let $D = \sqrt{d}/8 \geq 3$. By the Chernoff-Hoeffding bound (Lemma C.11), we get

$$\Pr[|N(S) \cap N(T)| > Dl] \leq \exp(-Dl\log(Dl/(p^2 l^2 me))) \leq \left(\frac{D}{p^2 lme}\right)^{-Dl}.$$

By the union bound over sets $S, T$ of size $l$,

$$q_l \leq \left(\frac{D}{p^2 lme}\right)^{-Dl}\left(\frac{en}{l}\right)^{2l} \leq (p^{2D}m^D l^{D-2} n^2)^l = \left(\frac{d^{2D}l^{D-2}n^2}{m^D}\right)^l.$$

So long as $l \leq m/d^3$ and $d \geq \log^2 n$, we have $d^{2D}l^{D-2}n^2/m^D \leq n^2/(m^2 d^{D-6}) \leq 1/m^2$ for large $n$. As a result, summing over $l$, we have

$$\sum_{l=1}^{m/d^3} q_l \leq \sum_{l=1}^{m/d^3} (1/m^2)^l \leq 2/m^2$$

as claimed. $\qquad\square$

Together, the above three lemmas immediately imply that the random matrix $M$ satisfies Assumption 1 with high probability.

**Theorem C.13.** *Let $n$ be an even positive integer that is sufficiently large. Let $m, d \in \mathbb{N}$ with $d \leq m \leq n$, and let $\epsilon \in (0,1)$. If $d \geq 32\epsilon^{-2}\log n$ and $d \geq \log^2 n$, then with probability at least $1 - 4/n - 2/m^2$, the random binary matrix $M \in \mathbb{R}^{m \times n}$ with i.i.d. entries $M_{ij} \sim Ber(d/m)$ satisfies Assumption 1 with sparsity parameter $k = m/d^3$ and error parameter $\epsilon$.*

Putting together Theorem C.13 and Corollary C.7 immediately gives the following.

**Theorem C.14.** *There are constants $N, C$ with the following property. Let $n, m \in \mathbb{N}$ with $n \geq N$. Let $p \in (0,1)$ satisfy $p \geq C(\log^2 n)/m$. Let $M$ be the $m \times n$ random matrix with independent entries $M_{ij} \sim Ber(p)$. Then with probability $1 - O(1/n) - O(1/m^2)$, it holds that for all $b \leq m/d^4$, the matrix $M$ is $(b, 2b, d^4 n^6, m/d^3)$-erasure-robust.*

### C.3 Properties of Final Construction

We can now prove the following existence result, which collects all the important properties of the matrix $M$ which we will use for our lower bound against Preconditioned Lasso: sparsity, quantitative density of the kernel, erasure-robustness, and eigenvalue bounds.

**Theorem C.15.** *Let $n \in \mathbb{N}$ be an even number larger than some absolute constant $n_0$. There is a density parameter $\tau = \Omega(n/\log^6 n)$, $\eta = O(n^6 \log^8 n)$, $b = \Omega(n/\log^8 n)$, and a matrix $M \in \mathbb{R}^{n/2 \times n}$ with the following properties:*

1. *The rows of $M$ are $O(\log^2 n)$-sparse*

2. *For any $x \in \mathbb{R}^n$ with $\mathrm{dist}(x, \ker M) \leq \|x\|_2 / (12\sqrt{n})$, and any $S \subseteq [n]$ with $|S| \leq \tau$, it holds that $\|x_{S^c}\|_2 \geq \|x\|_2 / (2\sqrt{n})$.*

3. *For any $B \subseteq [n/2]$ with $|B| \leq b$, there is a set $C \subseteq [n]$ with $|C| \leq 2|B|$, such that for any $x \in \mathbb{R}^n$, either*

   - $\|x_{C^c}\|_2 \leq \eta \|M_{B^c} x\|_\infty$, *or*
   - $|\mathrm{supp}(x)| \geq \tau$.

4. $\left\| M^T M \right\|_F \leq O(n \log n)$

5. *The smallest nonzero eigenvalue $\lambda$ of $\Theta$ satisfies $\lambda \geq \Omega(n^{-5/2})$.*

*Proof.* Let $\epsilon = 1/17$ and $d = \log^2 n$. Let $M \in \mathbb{R}^{n/2 \times n}$ be the random binary matrix with independent entries $M_{ij} \sim \mathrm{Ber}(2d/n)$. By Theorem C.13, $M$ satisfies Assumption 1 with $k = n/\log^6 n$ and error $\epsilon$, with probability at least $1 - 12/n$. Let $\tau = k$. Claim (1) follows from the degree bound condition.

To prove claim (2), let $x \in \mathbb{R}^n$ with $\mathrm{dist}(x, \ker M) \leq \|x\|_2 / (12\sqrt{n})$, and let $C \subseteq [n]$ with $|C| \leq k$. Let $y = \mathrm{Proj}_{\ker M} x$. Then $My = 0$, so $\|y_C\|_1 \leq \|y\|_1 / 3$ by Lemma C.8. Therefore

$$
\begin{aligned}
\|x_{C^c}\|_2 &\geq \|y_{C^c}\|_2 - \frac{1}{12\sqrt{n}} \|x\|_2 \\
&\geq \frac{1}{\sqrt{n}} \|y_{C^c}\|_1 - \frac{1}{12\sqrt{n}} \|x\|_2 \\
&\geq \frac{2}{3\sqrt{n}} \|y\|_1 - \frac{1}{12\sqrt{n}} \|x\|_2 \\
&\geq \frac{2}{3\sqrt{n}} \|y\|_2 - \frac{1}{12\sqrt{n}} \|x\|_2 \\
&\geq \frac{1}{2\sqrt{n}} \|x\|_2
\end{aligned}
$$

as desired.

Claim (3) follows from Corollary C.7 with $b = k/d$. Claim (4) follows from the degree bound assumption, and claim (5) holds with probability at least $1/2$ by results from random matrix theory (Theorem 1.1 in [4]) , and the observation that $\lambda = \sigma^2$, where $\sigma$ is the smallest singular value of $M^T$.

Thus, all claims hold with probability at least $1/2 - 12/n > 0$, so in particular the desired matrix $M$ exists. □

## D  Structure Lemma under Erasure-Robustness

In this section we show how erasure-robust sparse designs $M$ can be used to construct a covariance matrix $\Sigma$ so that preconditioners which are "compatible" with most rows of $M$ (i.e. succeed at recovery with non-trivial probability when the covariates are drawn from $N(0, \Sigma)$ and the signal is

the row of $M$) satisfy a useful structure lemma. Concretely, the covariance matrix we use to fool the preconditioned Lasso is given by

$$\tilde{\Sigma} := \tilde{\Theta}^{-1}, \qquad \tilde{\Theta} := \Theta + \epsilon I_n, \qquad \Theta = M^T M$$

where $\epsilon > 0$ is polynomially small, so that $\tilde{\Sigma}$ still has polynomial condition number. Ultimately, we will instantiate $M$ as the matrix constructed in the previous section, but for now we state our results in generality. We will show that if $M$ is erasure-robust and a preconditioner is compatible with most rows of $M$, then the columns of the preconditioner are either dense or have small magnitude outside a small set of coordinates. Moreover, we'll show that the number of dense columns must be linear.

### D.1 Failure on incompatible signals

To start with, we recall the weak compatibility coefficients $\alpha^{(1)}$ and $\beta^{(1)}$ from [32], which for any fixed covariance matrix and preconditioner, provide a simple necessary condition for the success of the (Preconditioned) Lasso. (We remind the reader that for fixed $w^*$, there also exists a sharp characterization of its failure/success due to Gordon's theorem (see e.g. [2]), though the exact characterization is not as easy to work with directly.)

**Definition 7** (Weak $S$-Preconditioned Compatibility Condition [32])**.** We say that

$$\alpha^{(1)}_{\Sigma,S,k} = \inf_{w \in B_0(k) \setminus \{0\}} \frac{\langle w, \Sigma w \rangle}{\|S^T w\|_1^2}$$

where we let $B_0(k)$ denotes the set of $k$-sparse vectors and

$$\beta^{(1)}_{\Sigma,S,m,k} = \sup\{\beta \in \mathbb{R} : \dim W_{\Sigma,S,\beta} \geq 2m\}$$

where

$$W_{\Sigma,S,\beta} = \left\{ w : \langle w, \Sigma w \rangle \geq \beta \|S^T w\|_1^2 \right\},$$

and $\dim W_{\Sigma,S,\beta}$ is defined as the largest dimension of any subspace contained in $W_{\Sigma,S,\beta}$.

*Remark* 1. Since we only deal with invertible covariance matrices $\Sigma$ in this paper, it always holds that $\alpha^{(1)}_{\Sigma,S,k} > 0$. Moreover, so long as $S$ is not identically zero, $\alpha^{(1)}_{\Sigma,S,k}$ is finite. We also assume that $S^T$ has trivial kernel, which implies that $\beta^{(1)}_{\Sigma,S,m,k}$ is finite. This assumption is essentially without loss of generality; see Section B.1 for discussion.

In [32], it shown that if the ratio $\beta^{(1)}_{\Sigma,S,m,k}/\alpha^{(1}_{\Sigma,S,m,k}$ exceeds a certain constant, then there is a $k$-sparse signal such that the $S$-preconditioned Lasso with $m$ samples fails with high probability. However, this signal depends on $S$ (specifically, it's the signal for which $\alpha^{(1)}_{\Sigma,S,k}$ achieves the infimum). Because we ultimately need to construct a signal distribution independent of $S$, we need a slightly more general statement which provides a condition under which a given signal causes $S$-preconditioned Lasso to fail. Such a statement is in fact implicit in [32]:

**Theorem D.1** ([32])**.** *Let $\Sigma \in \mathbb{R}^{n \times n}$ be positive-definite and let $S \in \mathbb{R}^{n \times s}$. Let $m, k \in \mathbb{N}$. If $w^* \in \mathbb{R}^n$ is a $k$-sparse signal with*

$$(w^*)^T \Sigma w^* < \frac{\beta^{(1)}_{\Sigma,S,m,k}}{18} \|S^T w^*\|_1^2,$$

*then the $S$-preconditioned Lasso exactly recovers $w^*$ with probability at most $\exp(-\Omega(m))$, from $m$ samples with independent covariates $X_1, \ldots, X_m \sim N(0, \Sigma)$ and noiseless responses $Y_i = \langle w^*, X_i \rangle$.*

*Proof.* The proof of this result is implicit in (Theorem 6.5, [32]) and we include it for the reader's convenience. For convenience of notation let $\beta = \beta^{(1)}_{\Sigma,S,m,k}$ and $\Theta = \Sigma^{-1}$. We want to show that with high probability, the $S$-preconditioned Lasso (1) fails to recover $w^*$, i.e.

$$w^* \notin \underset{w : Xw = Xw^*}{\operatorname{argmin}} \|S^T w\|_1$$

where $X$ has rows $X_1, \ldots, X_m \sim N(0, \Sigma)$. We know that

$$(w^*)^T \Sigma w^* = \alpha \left\| S^T w^* \right\|_1^2$$

for some $\alpha < \beta/18$.

By definition of $\beta$, there is a subspace $U \subseteq \mathbb{R}^n$ of dimension $2m$ such that $w^T \Sigma w \geq \beta \left\| S^T w \right\|_1^2$ for all $w \in U$. Let $v_1, \ldots, v_{2m} \in U$ form an orthonormal basis for $U$, and let $V \in \mathbb{R}^{n \times 2m}$ be the matrix with columns $v_1, \ldots, v_{2m}$.

We construct $v \in \mathbb{R}^n$ to satisfy $Xw^* = Xv$ and $\left\| S^T v \right\|_1 < \left\| S^T w^* \right\|_1$ as follows. Let $\Gamma = V^T \Sigma V \in \mathbb{R}^{2m \times 2m}$. The columns of $V$ have no linear dependencies, and $\Sigma$ is symmetric positive-definite, so $\Gamma$ is symmetric positive-definite. Thus, there is an invertible matrix $N \in \mathbb{R}^{2m \times 2m}$ such that $\Gamma = N^T N$. Define

$$c = N^{-1} (XVN^{-1})^\dagger Xw^* \in \mathbb{R}^{2m}$$

and define $v = Vc \in \mathbb{R}^n$. By construction we have $v \in U$, so

$$v^T \Sigma v \geq \beta \left\| S^T v \right\|_1^2. \tag{2}$$

Second, note that

$$\mathbb{E}[(XVN^{-1})^T (XVN^{-1})] = m(N^{-1})^T V^T \Sigma V N^{-1} = m(N^{-1})^T \Gamma N^{-1} = m I_{2m}.$$

Moreover, the rows of $XVN^{-1}$ are independent and Gaussian. So in fact $XVN^{-1}$ has i.i.d. $N(0,1)$ entries. Thus, with probability $1 - \exp(-\Omega(m))$, we have $\sigma_{\min}((XVN^{-1})^T) \geq \sqrt{m}/3$ since the dimensions of $(XVN^{-1})^T$ are $2m \times m$ (by Theorem B.1). Hence, $\sigma_{\max}((XVN^{-1})^\dagger) \leq 3/\sqrt{m}$. We can conclude that

$$v^T \Sigma v = c^T N^T N c = (w^*)^T X^T (XVN^{-1})^{\dagger T} (XVN^{-1})^\dagger Xw^* \leq (9/m)(w^*)^T X^T Xw^*. \tag{3}$$

We can now check that $\left\| S^T v \right\|_1 < \left\| S^T w \right\|_1$. Indeed, by Theorem B.2, $(w^*)^T X^T Xw^* \leq 2m(w^*)^T \Sigma w^*$ with probability at least $1 - \exp(-\Omega(m))$, so

$$
\begin{aligned}
\left\| S^T v \right\|_1 &\leq \sqrt{\frac{1}{\beta} v^T \Sigma v} \\
&\leq \sqrt{\frac{9}{m\beta} (w^*)^T X^T Xw^*} \\
&\leq \sqrt{\frac{18}{\beta} (w^*)^T \Sigma w^*} \\
&\leq \sqrt{\frac{18\alpha}{\beta}} \left\| S^T w^* \right\|_1
\end{aligned}
$$

which produces the desired inequality as long as $\beta/\alpha > 18$.

Finally, since $XVN^{-1}$ is rank-$m$ with probability 1, we have $(XVN^{-1})(XVN^{-1})^\dagger = I_m$, and thus

$$Xv = XVN^{-1}(XVN^{-1})^\dagger Xw^* = Xw^* \tag{4}$$

as desired. $\qquad \square$

## D.2 Structure Lemma for compatible preconditioners

We now consider the case where the preconditioner does a good job of preconditioning the rows $M_i$, so that Lasso can succeed with non-trivial probability for most such signals. Then very informally, one would think this forces $\Sigma \approx SS^T$, which by the construction of $\Sigma$ will force $S$ to be dense, and hence the preconditioner will fail on basis vectors.

The following Lemma starts to make this intuition correct and precise. It shows that if the preconditioner is compatible with most of the rows of $M$, then the "sparse part" of the preconditioner $S$ is not too large. The "sparse part" is actually given not just by removing the dense rows of $S^T$ but also by removing a small number $b'$ of columns of $S^T$: this is inevitable because the assumption that the preconditioner is compatible with *most* of the rows of $M$ cannot imply something about *all* of the columns of $S^T$, and this is where erasure-robustness is crucially used.

**Lemma D.2** (Structure Lemma). *Let $M \in \mathbb{R}^{n-r \times n}$, and define $\Theta = M^T M$. Let $\lambda$ be the smallest nonzero eigenvalue of $\Theta$. Let $\epsilon > 0$, and let $\tilde{\Theta} = \Theta + \epsilon I$. Let $k, m, \alpha, \tau, b, b', \eta > 0$. Suppose that $M$ satisfies $(b, b', \eta, \tau)$-erasure-robustness (Definition 4).*

*Let $S \in \mathbb{R}^{n \times s}$. Suppose that*

$$\Pr_{i \in [n-r]} \left[ M_i^T \tilde{\Theta}^{-1} M_i < \alpha \left\| S^T M_i \right\|_1^2 \right] \leq \frac{b}{n-r}$$

*and define $\gamma = \beta^{(1)}_{\tilde{\Theta}^{-1}, S, k, m} / \alpha$. Let*

$$D := \{ i \in [s] : \|(S^T)_i\|_0 \geq \tau \}$$

*be the set of $\tau$-dense columns of $S$ (rows of $S^T$). There exists a subset of row indices $C \subseteq [n]$ with $|C| \leq b'$ such that the submatrix $S_{C^c D^c}$ satisfies*

$$\sum_{j \in D^c} \|S_{C^c j}\|_2 \leq \frac{n^{3/2} \eta \|M\|_F}{\sqrt{\lambda \alpha}}.$$

*Proof.* Let $B \subseteq [n-r]$ be the set of $i \in [n-r]$ such that $M_i^T \tilde{\Theta}^{-1} M_i < \alpha \left\| S^T M_i \right\|_1^2$. By assumption, $|B| \leq b$; let $C := C_B \subseteq [n]$ be the set guaranteed by erasure-robustness, which indeed satisfies $|C| \leq b'$. For any $i \notin B$, we have

$$\left\| S^T M_i \right\|_1^2 \leq \frac{1}{\alpha} M_i^T \tilde{\Theta}^{-1} M_i \leq \frac{\|M_i\|_2^2}{\lambda \alpha},$$

where the last inequality is because $M_i \in \text{span}\,\Theta$. Expanding the $\ell_1$ norm as a sum, and summing $\left\| S^T M_i \right\|_1$ over $i \notin B$, we have

$$\sum_{j \in [s]} \sum_{i \notin B} |\langle (S^T)_j, M_i \rangle| \leq \frac{n \sum_{i \notin B} \|M_i\|_2}{\sqrt{\lambda \alpha}} \leq \frac{n^{3/2} \|M\|_F}{\sqrt{\lambda \alpha}}.$$

Thus,

$$\sum_{j \in [s]} \left\| M_{B^c} (S^T)_j \right\|_\infty \leq \sum_{j \in [s]} \left\| M_{B^c} (S^T)_j \right\|_1 \leq \frac{n^{3/2} \|M\|_F}{\sqrt{\lambda \alpha}}.$$

As specified in the theorem statement, let $D \subseteq [s]$ be the set of $i \in [s]$ such that $\|(S^T)_i\|_0 \geq \tau$. Then for any $i \in D^c$, we have that $\left\| (S^T)_{iC^c} \right\|_2 \leq \eta \left\| M_{B^c} (S^T)_i \right\|_\infty$. Thus,

$$\sum_{j \in D^c} \left\| (S^T)_{jC^c} \right\|_2 \leq \eta \sum_{j \in D^c} \left\| M_{B^c} (S^T)_j \right\|_\infty \leq \frac{n^{3/2} \eta \|M\|_F}{\sqrt{\lambda \alpha}}$$

as claimed. $\qquad \square$

As a consequence of the previous lemma, we can show that if the compatibility ratio is small then $S$ must have many dense columns ($S^T$ has many dense rows).

**Lemma D.3.** *In the setting of Lemma D.2, suppose that $1/\sqrt{\beta^{(1)}_{\tilde{\Theta}^{-1}, S, k, m} \epsilon} > n^{3/2} \eta \|M\|_F / \sqrt{\lambda \alpha}$. Then $|D| > n - r - b' - 2m$.*

*Proof.* Suppose for contradiction that $|D| \leq n - r - b' - 2m$. Pick some $\beta' > \beta^{(1)}_{\tilde{\Theta}^{-1}, S, k, m}$ sufficiently small that $1/\sqrt{\beta' \epsilon} > n^{3/2} \eta \|M\|_F / \sqrt{\lambda \alpha}$. Define

$$W = \ker(S_D^T) \cap \ker(\Theta) \cap \text{span}\{e_i : i \in C^c\} \subseteq \mathbb{R}^n.$$

Then

$$\dim(W) \geq n - |D| - r - |C| \geq 2m.$$

For any $w \in W$, we have

$$w^T \tilde{\Theta}^{-1} w \geq \epsilon^{-1} \|\mathrm{Proj}_{\ker \Theta} w\|_2^2 = \epsilon^{-1} \|w\|_2^2$$

since $w \in \ker \Theta$. On the other hand,

$$\|S^T w\|_1 = \sum_{j \in D^c} |\langle (S^T)_j, w \rangle| \qquad (w \in \ker(S_D^T))$$

$$= \sum_{j \in D^c} |\langle (S^T)_{jC^c}, w_{C^c} \rangle| \qquad (\mathrm{supp}(w) \subseteq C^c)$$

$$\leq \sum_{j \in D^c} \|(S^T)_{jC^c}\|_2 \|w\|_2 \qquad (\text{Cauchy-Schwarz})$$

$$\leq \frac{n^{3/2} \eta \|M\|_F}{\sqrt{\lambda \alpha}} \|w\|_2 . \qquad (\text{Lemma D.2})$$

As a consequence, by choice of $\beta'$, we have $w^T \tilde{\Theta}^{-1} w \geq \beta' \|S^T w\|_1^2$. Therefore $W \subseteq W_{\tilde{\Theta}^{-1}, S, \beta'}$, contradicting the definition of $\beta_{\tilde{\Theta}^{-1}, S, k, m}$. $\qquad \square$

## E  Failure of the Preconditioned Lasso

In this section, we prove Theorems 2.1 and 2.2. We start by proving our lower bound against invertible preconditioners. We then prove a key projection lemma and use it to prove the lower bound against rectangular preconditioners.

### E.1  Invertible Preconditioners

To construct a signal distribution which fails invertible preconditioners with probability $1 - o(1)$, we'll need to amplify the failure probability by adding together multiple signals. The following lemma formalizes why this works in certain cases: a random combination of vectors where at least one of them has large $\ell_1$ norm is very unlikely to have small $\ell_1$ norm.

**Lemma E.1.** *Let $\mathcal{D}$ be a continuous distribution on $\mathbb{R}$ with density upper bounded by $1/2$. Let $v_1, \ldots, v_t \in \mathbb{R}^n$ and let $Z_1, \ldots, Z_t \sim \mathcal{D}$ be independent random variables. Then*

$$\Pr\left[ \left\| \sum_{i=1}^t Z_i v_i \right\|_1 < \delta \max_{i \in [t]} \|v_i\|_1 \right] \leq \delta.$$

*Proof.* Without loss of generality assume that $\|v_t\|_1 = \max_{i \in [t]} \|v_i\|_1$. Condition on $Z_1, \ldots, Z_{t-1}$ and define

$$f(z) = \left\| \sum_{i=1}^{t-1} Z_i v_i + z v_t \right\|_1 .$$

For any $z, z'$ with $f(z) \leq \delta \|v_t\|_1$ and $f(z') \leq \delta \|v_t\|_1$ we have by the triangle inequality that $\|z v_t - z' v_t\|_1 \leq 2\delta \|v_t\|_1$, so $|z - z'| \leq 2\delta$. Since the density of $\mathcal{D}$ is upper bounded by $1/2$, it follows that

$$\Pr[f(Z_t) < \delta \|v_t\|_1] \leq \frac{2\delta}{2} = \delta$$

which proves the claim. $\qquad \square$

We know show that if $M$ is a sparse compressive erasure-robust design matrix (e.g. as we constructed previously), then the covariance matrix $\Sigma = (M^T M + \epsilon I)^{-1}$ together with an appropriate signal distribution describes a hard distribution family for (invertibly) Preconditioned Lasso.

**Theorem E.2.** *Let $n, r \in \mathbb{N}$. Let $M \in \mathbb{R}^{n-r \times n}$ and $\epsilon > 0$. Define $\Theta = M^T M$ and $\tilde{\Theta} = \Theta + \epsilon I$. Let $\lambda$ be the smallest nonzero eigenvalue of $\Theta$. Let $k, m, \alpha, \tau, \eta, b, b', t > 0$ and suppose that $M$ satisfies $(b, b', \eta, \tau)$-erasure-robustness and has $k$-sparse rows.*

*Suppose $k > 2(n/\tau) \log(n)$ and $n - r - b' \geq 3m$. There is a distribution $\mathcal{D}$ on $k(t+1)$-sparse signals in $\mathbb{R}^n$ with the following property.*

*Let $S \in \mathbb{R}^{n \times n}$ be invertible. Suppose that*

$$9n^{105/2}\eta \left\|M\right\|_F \sqrt{\epsilon/\lambda} < 1. \tag{5}$$

*Then with probability at least $1 - \exp(-tb/(n-r)) - 1/n^{50}$ over true signals $w^* \sim \mathcal{D}$, it holds that $S$-preconditioned Lasso fails with probability $1 - \exp(-\Omega(m))$ over independent samples $X_1, \ldots, X_m \sim N(0, \tilde{\Theta}^{-1})$: that is, $w^*$ is not a unique minimizer of $\left\|S^T w\right\|_1$ subject to $Xw = Xw^*$.*

*Proof.* Let $\mathcal{D}$ be the signal distribution where we draw independent and uniformly random indices $R_1, \ldots, R_t \in [n-r]$ as well as independent $Z_0, Z_1, \ldots, Z_t \sim \text{Unif}([-1,1])$, and draw $\tilde{w}$ with uniformly random $k$-sparse support and entries $\text{Unif}([-1,1])$ on that support, and set the signal to be

$$w^* = Z_0 \sqrt{\epsilon}\tilde{w} + \sum_{i=1}^{t} \frac{Z_i M_{R_i}}{\sqrt{M_{R_i}^T \tilde{\Theta}^{-1} M_{R_i}}}.$$

Pick any invertible $S \in \mathbb{R}^{n \times n}$. Define $\alpha = \beta^{(1)}_{\tilde{\Theta}^{-1}, S, k, m}/(72n^{102})$. We distinguish two cases.

**Case I: incompatible preconditioner.** On the one hand, suppose that

$$\Pr_{i \in [n-r]} \left[ M_i \tilde{\Theta}^{-1} M_i < \alpha \left\|S^T M_i\right\|_1^2 \right] \geq \frac{b}{n-r}.$$

Then with probability at least

$$\left(1 - (1 - b/(n-r))^t\right) \geq \left(1 - e^{-tb/(n-r)}\right)$$

over the row indices $R_1, \ldots, R_t$, there is some $R_i$ with

$$\left\|S^T M_{R_i}\right\|_1^2 > \frac{1}{\alpha} M_{R_i}^T \tilde{\Theta}^{-1} M_{R_i}.$$

Under this event, we have

$$\max_{i \in [t]} \frac{\left\|S^T M_{R_i}\right\|_1}{\sqrt{M_{R_i}^T \tilde{\Theta}^{-1} M_{R_i}}} > \frac{1}{\sqrt{\alpha}},$$

so by Lemma E.1, it holds with probability at least $1 - 1/n^{50}$ over $Z_1, \ldots, Z_t$ that $\left\|S^T w^*\right\|_1 \geq \frac{1}{n^{50}\sqrt{\alpha}}$. But by the triangle inequality, we have $\sqrt{(w^*)^T \tilde{\Theta}^{-1} w^*} \leq t + \sqrt{k} \leq 2n$ (since $\tilde{w}^T \tilde{\Theta}^{-1} \tilde{w} \leq \epsilon^{-1} \left\|\tilde{w}\right\|_2^2$). Thus,

$$\Pr_{w^* \sim \mathcal{D}} \left[ \left\|S^T w^*\right\|_1^2 > \frac{1}{4n^{102}\alpha}(w^*)^T \tilde{\Theta}^{-1} w^* \right] \geq \left(1 - e^{-tb/(n-r)}\right)\left(1 - \frac{1}{n^{50}}\right).$$

Moreover, for such $w^*$, by choice of $\alpha$ and by Theorem D.1, the $S$-preconditioned Lasso recovers $w^*$ with probability at most $\exp(-\Omega(m))$, from $m$ independent samples $X_1, \ldots, X_m \sim N(0, \tilde{\Theta}^{-1})$.

**Case II: compatible preconditioner.** On the other hand, suppose that

$$\Pr_{i \in [n-r]} \left[ M_i \tilde{\Theta}^{-1} M_i \geq \alpha \left\|S^T M_i\right\|_1^2 \right] \geq \frac{b}{n-r}.$$

By choice of $\alpha$ and the theorem assumptions, we know that $1/\sqrt{\beta^{(1)}_{\tilde{\Theta}^{-1}, S, k, m}\epsilon} > n^{3/2}\eta \left\|M\right\|_F /\sqrt{\lambda\alpha}$. So we can apply Lemma D.2 and Lemma D.3: there is a set $D \subseteq [s]$ satisfying $|D| > n - r - b' - 2m$, and $\left\|(S^T)_j\right\|_0 \geq \tau$ for all $j \in D$. Let $U = \text{supp}(S^T w^*)$. Then since the support of $\tilde{w}$ is uniformly random of size $k \geq 2(n/\tau)\log(n)$, we know that $U \supseteq \text{supp}(S^T \tilde{w}) \supseteq D$ with probability at least $1 - 1/n$. Let $V = \{d \in \mathbb{R}^n : \text{supp}(S^T d) \subseteq U\}$ and let $z = \text{sign}(S^T w^*)$. We claim that there is some $d \in V$ with $Xd = 0$ but $\langle d, Sz \rangle \neq 0$. Indeed, since $S^T$ is invertible, it suffices to show that there is a vector $f \in \mathbb{R}^n$ supported on $U$, such that $\langle S^{-1}X_i, f \rangle = 0$ for all

$i \in [m]$, but $\langle z, f \rangle \neq 0$. This holds because $z$ is with probability 1 outside the span of the vectors $\{(S^{-1}X_i)_U : i \in [m]\}$, and the number of degrees of freedom is $|U| \geq |D| > n - r - b' - 2m \geq m$. We conclude that the desired direction of improvement $d$ exists, so $w^*$ is not a minimizer of the $S$-preconditioned Lasso.

$\square$

*Remark* 2. The term $1/n^{50}$ in the probability of failure in Theorem E.2 and Theorem E.3 can be replaced by $1/n^\ell$ for any particular $\ell$ if we modify the left hand side of (5) accordingly. All that happens if we pick a larger $\ell$ is that to satisfy (5), we need to pick a correspondingly (polynomially) smaller $\epsilon$ which means that the covariance matrix of the data, $\Sigma = \tilde{\Theta}^{-1}$, becomes more ill-conditioned. We stated the results with $\ell = 50$ only to simplify the statements.

Finally, we instantiate the above theorem with the parameters of the design matrix $M$ we constructed in Theorem C.15.

**Theorem E.3.** *Let $n \in \mathbb{N}$ be sufficiently large. There is a matrix $\Sigma \in \mathbb{R}^{n \times n}$ with condition number $\mathrm{poly}(n)$ such that the following holds. Let $k \in \mathbb{N}$ with $k \geq \log^8 n$. There is a distribution $\mathcal{D}$ over $O(k\log^9 n)$-sparse signals such that for any positive integer $m \leq n/7$ and any invertible preconditioner $S \in \mathbb{R}^{n \times n}$, with probability at least $1 - O(1/n^{50})$ over $w^* \sim \mathcal{D}$, the $S$-preconditioned Lasso recovers $w^*$ with probability at most $\exp(-\Omega(m))$ from $m$ independent samples $X_1, \ldots, X_m \sim N(0, \Sigma)$ with noiseless responses $Y_i = \langle X_i, w^* \rangle$.*

*Proof.* Let $\Theta$ be the matrix guaranteed by Theorem C.15. We check the conditions of Theorem E.2. First, $\dim \ker(\Theta) \geq n/2$. Second, $\lambda = \Omega(n^{-5/2})$. We can take $b = n/(\log^8 n)$, $b' = 2n/(\log^8 n)$, $\eta = n^6 \log^8 n$, and $\tau = \Omega(n/\log^7 n)$. We have $\|M\|_F \leq O(\sqrt{n \log n})$. Thus, we can take $\epsilon = \Omega(n^{-111})$. We know that the rows of $M$ are $k$-sparse. Let $t = 2\log^9 n$.

Applying Theorem E.2, there is a distribution $\mathcal{D}$ over $O(k\log^9 n)$-sparse signals such that for any invertible preconditioner $S \in \mathbb{R}^{n \times n}$, with probability at least $1 - O(1/n^{50})$ over $w^* \sim \mathcal{D}$, the $S$-preconditioned Lasso recovers $w^*$ uniquely with probability $\exp(-\Omega(m))$ from $m$ samples, so long as $m \leq n/7$ (so that $n - r - b' \geq 3m$). $\square$

## E.2 Projection Lemma

To extend our lower bound to rectangular preconditioners, we need Lemma E.6, a projection lemma generalizing an analogous result from [32]. To prove it, we recall two lemmas which are essentially taken from [32]; the second of these is the original projection lemma.

First, recall that our covariance matrix has the form $\tilde{\Sigma} = \tilde{\Theta}^{-1}$, where $\tilde{\Theta} = \Theta + \epsilon I$ for some PSD matrix $\Theta$. The following lemma establishes that if $\epsilon$ is sufficiently small relative to the smallest nonzero eigenvalue of $\Theta$, then the row span of the design matrix $X$ is nearly orthogonal to all but the top eigenspace of the covariance $\tilde{\Sigma} = \left(\tilde{\Theta}\right)^{-1}$ (i.e. the kernel of $\Theta$). In other words, by taking $\epsilon$ small enough the top eigenspace dominates, as expected.

**Lemma E.4** ([32]). *Let $\Theta \in \mathbb{R}^{n \times n}$ be a PSD matrix with minimum nonzero eigenvalue $\lambda$. Let $\epsilon, m > 0$ and let $\tilde{\Theta} = \Theta + \epsilon I$. Let $X_1, \ldots, X_m \sim N(0, \tilde{\Theta}^{-1})$. If $r := \dim \ker \Theta > 2m$, then with probability at least $1 - \exp(-\Omega(m))$ it holds that for all $a \in \mathbb{R}^m$,*

$$\left\|\mathrm{Proj}_{\mathrm{span}\,\Theta}\, X^T a\right\|_2 \leq C\sqrt{\frac{n\epsilon}{\lambda}} \left\|X^T a\right\|_2$$

*where $C$ is an absolute constant, and where $X : m \times n$ is the matrix with rows $X_1, \ldots, X_m$.*

*Proof.* The proof of this result is essentially contained in [32], and we repeat it to make this paper self-contained. The statement of the lemma is basis-independent (e.g. does not depend on sparsity of $\Theta$ or $a$), so we can assume without loss of generality that $\Theta$ is diagonal. Then $\tilde{\Theta}^{-1}$ is diagonal, and we can choose a basis ordering such that the first $r = \dim \ker \Theta$ diagonal entries are each $\epsilon^{-1}$.

Next, note that $(X^T)_{[r]}$ is a $r \times m$ matrix with i.i.d. $N(0, \epsilon^{-1})$ entries, so $\sigma_{\min}((X^T)_{[r]}) \geq c\epsilon^{-1/2}\sqrt{r}$ with probability at least $1 - \exp(-\Omega(m))$, for some constant $c > 0$ (by Theorem B.1). On the other

hand, since the entries of $\tilde{\Theta}^{-1}_{[r]^c,[r]^c}$ are bounded by $1/\lambda$, we also have $\sigma_{\max}((X^T)_{[r]^c}) \leq C\sqrt{n/\lambda}$ with probability at least $1 - \exp(-\Omega(n))$, for some constant $C$. This means that for any $u \in \mathbb{R}^m$,

$$\left\|(X^T u)_{[r]^c}\right\|_2 \leq C\sqrt{\frac{n}{\lambda}}\left\|u\right\|_2 \leq \frac{C\sqrt{n\epsilon}}{c\sqrt{\lambda r}}\left\|(X^T u)_{[r]}\right\|_2 \leq \frac{C}{c}\sqrt{\frac{n\epsilon}{\lambda}}\left\|X^T u\right\|_2.$$

But $(X^T u)_{[r]^c}$ is precisely the projection of $X^T u$ onto $\mathrm{span}\,\Theta$. So this proves the lemma. $\qquad\square$

The next lemma establishes that if the top eigenspace of the covariance $\tilde{\Sigma} = \left(\tilde{\Theta}\right)^{-1}$ has a dimension significantly larger than the number of samples, then the row span of the design matrix $X$ is unlikely to align with any particular direction $v$ (i.e. the projection of $v$ onto the null space has large norm). Informally, this is because the worst-case $v$ to consider would be a vector in the top eigenspace (since a lot of the energy of the samples is in this space), and a random lower-dimensional subspace of the top eigenspace (corresponding to the samples) is not likely to contain any particular direction.

**Lemma E.5** (Lemma 7.4 in [32]). *Let $\Theta \in \mathbb{R}^{n \times n}$ be a PSD matrix with minimum nonzero eigenvalue $\lambda$. Let $\epsilon, m > 0$ and let $\tilde{\Theta} = \Theta + \epsilon I$. Let $X_1, \ldots, X_m \sim N(0, \tilde{\Theta}^{-1})$. If $\epsilon \leq c\lambda/n$ for a sufficiently small absolute constant $c > 0$, and $r := \dim \ker \Theta > 2m$, then for any fixed $v \in \mathbb{R}^n$, we have*

$$\Pr_{X_1,\ldots,X_m}[v^T(I - P)v \geq (v^T v)/8] \geq 1 - \frac{4m}{3r} - \exp(-\Omega(m)),$$

*where $P = X^T(XX^T)^{-1}X$ is the projection map onto $\mathrm{span}\{X_1, \ldots, X_m\}$, and where $X : m \times n$ is the matrix with rows $X_1, \ldots, X_m$. As an equivalent statement, it holds with probability at least $1 - (4m)/(3r) - \exp(-\Omega(m))$ that*

$$\inf_{a \in \mathbb{R}^m}\left\|v - X^T a\right\|_2 \geq \frac{1}{2\sqrt{2}}\left\|v\right\|_2.$$

*Proof.* The proof of this result is essentially contained in [32], and we repeat it to make this paper self-contained.

The statement of the lemma is basis-independent (e.g. does not depend on sparsity of $\Theta$ or $v$), so we can assume without loss of generality that $\Theta$ is diagonal. Then $\tilde{\Theta}^{-1}$ is diagonal, and we can choose a basis ordering such that the first $r = \dim \ker \Theta$ diagonal entries are each $\epsilon^{-1}$. Let $w = v_{[r]}$ be the first $r$ coordinates of $v$. For $i \in [m]$ let $Y_i = (X_i)_{[r]}$ be the first $r$ coordinates of $X_i$. Then $Y_1, \ldots, Y_m$ are i.i.d. $N(0, \epsilon^{-1}I_r)$. So if $P_Y = Y^T(YY^T)^{-1}Y$, then $P_Y$ is projection onto an isotropically random dimension-$m$ subspace of $\mathbb{R}^r$. Hence,

$$\mathbb{E}\left\|P_Y w\right\|_2^2 = \frac{m}{r}\left\|w\right\|_2^2.$$

With probability at least $1 - 4m/(3r)$ we have $\left\|P_Y w\right\|_2^2 \leq \frac{3}{4}\left\|w\right\|_2^2$. So $\left\|w - P_Y w\right\|_2^2 \geq \left\|w\right\|_2^2/4$. Now $\left\|w - P_Y w\right\|_2$ is the distance from $w$ to the subspace $\mathrm{span}\{Y_1, \ldots, Y_m\}$. For any vector in $\mathrm{span}\{X_1, \ldots, X_m\}$, its first $r$ coordinates lie in $\mathrm{span}\{Y_1, \ldots, Y_m\}$, so the distance to $v$ must be at least $\left\|w - P_Y w\right\|_2$. Thus,

$$\left\|v - Pv\right\|_2^2 \geq \left\|w - P_Y w\right\|_2^2 \geq \frac{1}{4}\left\|w\right\|_2^2. \tag{6}$$

Next, note that $(X^T)_{[r]}$ is a $r \times m$ matrix with i.i.d. $N(0, \epsilon^{-1})$ entries, so $\sigma_{\min}((X^T)_{[r]}) \geq c\epsilon^{-1/2}\sqrt{r}$ with probability at least $1 - \exp(-\Omega(m))$, for some constant $c > 0$ (by Theorem B.1). On the other hand, since the entries of $\tilde{\Theta}^{-1}_{[r]^c,[r]^c}$ are bounded by $1/\lambda$, we also have $\sigma_{\max}((X^T)_{[r]^c}) \leq C\sqrt{n/\lambda}$ with probability at least $1 - \exp(-\Omega(n))$, for some constant $C$. This means that for any $u \in \mathbb{R}^m$,

$$\left\|(X^T u)_{[r]^c}\right\|_2 \leq C\sqrt{\frac{n}{\lambda}}\left\|u\right\|_2 \leq \frac{C\sqrt{n\epsilon}}{c\sqrt{\lambda r}}\left\|(X^T u)_{[r]}\right\|_2.$$

By assumption, $\epsilon \leq (c/4C)^2\lambda r/n$, so that $\left\|(X^T u)_{[r]^c}\right\|_2 \leq \left\|(X^T u)_{[r]}\right\|_2/4$. Now $Pv$ lies in the span of $X_1, \ldots, X_m$, so there is some $u \in \mathbb{R}^m$ with $Pv = X^T u$. This means that

$$\left\|(Pv)_{[r]^c}\right\|_2 \leq \frac{1}{4}\left\|Pv\right\|_2 \leq \frac{1}{4}\left\|v\right\|_2.$$

So
$$\|v - Pv\|_2 \geq \left\|(v - Pv)_{[r]^c}\right\|_2 \geq \left\|v_{[r]^c}\right\|_2 - \frac{1}{4}\|v\|_2.$$
Together with Equation 6, which states that $\|v - Pv\|_2 \geq \frac{1}{2}\left\|v_{[r]}\right\|_2$, we get that $\|v - Pv\|_2^2 \geq \frac{1}{8}\|v\|_2^2$. $\qquad\qquad\square$

Finally, we extend the previous lemma to show that projection of a fixed direction onto the null space of the covariates has lower bounded norm even when restricted to a set of coordinates $P$ with sparse complement. We need an extra condition, that $\ker(\Theta)$ is quantitatively dense (Definition 6), meaning that there is no approximate solution $u$ to the equation $u_P = 0$ near the top eigenspace of $\tilde{\Sigma} = \tilde{\Theta}^{-1}$. Then we show that under the assumptions of the previous two lemmas, for any particular vector $u_P$ supported in $P$, the equation $u_P = (X^T a)_P$ is unlikely to have an approximate solutions $a$. The proof is by contradiction: if the conclusion is false, then there is a positive probability that if we sample two independent design $X, \bar{X}$ that: (1) there are approximate solutions $a, \bar{a}$ to $u_P = (X^T a)_P$ and $u_P = ((\bar{X})^T \bar{a})_P$, (2) by Lemma E.4 both approximate solutions are near the top eigenspace, and (3) by Lemma E.5 these two solutions are not well-aligned; combining (1-3) gives that $(X^T a - (\bar{X})^T \bar{a})_P$ is an approximate solution to $(u_P) = 0$ near the top eigenspace, which is impossible.

**Lemma E.6.** *There are absolute constants $c, C > 0$ such that the following holds. Let $\Theta \in \mathbb{R}^{n \times n}$ be a PSD matrix with minimum nonzero eigenvalue $\lambda$. Let $\epsilon > 0$ and $\delta, \eta, \tau > 0$ and let $\tilde{\Theta} = \Theta + \epsilon I$. Let $m \geq C \log n$. Suppose that $\epsilon \leq c\delta^2\lambda/n$, and $r := \dim \ker \Theta > 2m$, and $\ker(\Theta)$ is $(\delta, \eta, \tau)$-quantitatively dense (Definition 6). Fix $u \in \mathbb{R}^n$ and $P \subseteq [n]$ with $|P| \geq n - \tau$. Then with probability at least $p = 1 - (4m)/(3r) - \exp(-\Omega(m))$ over independent $X_1, \ldots, X_m \sim N(0, \tilde{\Theta}^{-1})$, it holds that*
$$\inf_{a \in \mathbb{R}^m} \left\|u_P - (X^T a)_P\right\|_2 \geq \frac{\eta}{32}\|u_P\|_2.$$

*Proof.* Suppose that the claim is false. Then using that $r = \dim \ker \Theta \leq n$ we have
$$\Pr_X \left[\exists a \in \mathbb{R}^m : \left\|(u - X^T a)_P\right\|_2 < (\eta/32)\|u_P\|_2\right] > \frac{4m}{3r} \geq \frac{1}{n}.$$
By Lemma E.4, we also have that
$$\Pr_X \left[\forall a \in \mathbb{R}^m : \left\|\text{Proj}_{\text{span}\,\Theta}\, X^T a\right\|_2 \leq C\sqrt{\frac{n\epsilon}{\lambda}}\left\|X^T a\right\|_2\right] \geq 1 - \exp(-\Omega(m)).$$
Since $m \geq C \log n$, for a sufficiently large constant $C$ the latter probability must exceed $1 - 1/n$. So with positive probability, these two events occur simultaneously. Hence, there exists some deterministic $\bar{X} \in \mathbb{R}^{m \times n}$ such that both events occur. Let $\bar{a}$ be the witness of the first event for $\bar{X}$, and define $v = \bar{X}^T \bar{a}$. Then the following two equations hold:
$$\|(u - v)_P\|_2 < \frac{\eta}{32}\|u_P\|_2 \tag{7}$$
$$\text{dist}(v, \ker \Theta) \leq C\sqrt{\frac{n\epsilon}{\lambda}}\|v\|_2 \tag{8}$$

Now suppose that the claim of Lemma E.4 holds (for the original samples $X_1, \ldots, X_m$). Also suppose that the claim of Lemma E.5 holds for vector $v$ (and samples $X_1, \ldots, X_m$), i.e. $v^T(I - P)v \geq (v^T v)/8$: or equivalently, the following inequality holds for arbitrary vectors $a \in \mathbb{R}^m$:
$$\left\|v - X^T a\right\|_2 \geq \frac{1}{2\sqrt{2}}\|v\|_2. \tag{9}$$
By Lemmas E.4 and E.5, we may assume that both claims hold with probability $p := 1 - (4m)/(3r) - \exp(-\Omega(m))$. Fix $a \in \mathbb{R}^m$. Then
$$\text{dist}(X^T a, \ker \Theta) \leq C\sqrt{\frac{n\epsilon}{\lambda}}\left\|X^T a\right\|_2 \qquad\qquad \text{(by claim of Lemma E.4)}$$
$$\leq C\sqrt{\frac{n\epsilon}{\lambda}}(\left\|v - X^T a\right\|_2 + \|v\|_2)$$
$$\leq 4C\sqrt{\frac{n\epsilon}{\lambda}}\left\|v - X^T a\right\|_2 \qquad\qquad \text{(by Equation 9)}$$

By Equation 8 and Equation 9,

$$\operatorname{dist}(v, \ker \Theta) \leq 3C \sqrt{\frac{n\epsilon}{\lambda}} \left\| v - X^T a \right\|_2.$$

Thus, by the triangle inequality,

$$\operatorname{dist}(v - X^T a, \ker \Theta) \leq 7C \sqrt{\frac{n\epsilon}{\lambda}} \left\| v - X^T a \right\|_2.$$

By assumption, $7C\sqrt{(n\epsilon)/\lambda} \leq \delta$. So because $\ker(\Theta)$ is $(\delta, \eta, \tau)$-quantitatively dense and $|P| \geq n - \tau$, and by Equation 9,

$$\left\| (v - X^T a)_P \right\|_2 \geq \eta \left\| v - X^T a \right\|_2 \geq \frac{\eta}{2\sqrt{2}} \left\| v \right\|_2.$$

Finally, we convert this into a bound on $(u - X^T a)_P$, repeatedly using the fact that vectors $u$ and $v$ are close on $P$ (Equation 7):

$$
\begin{aligned}
\left\| (u - X^T a)_P \right\|_2 &\geq \left\| (v - X^T a)_P \right\|_2 - \left\| (u - v)_P \right\|_2 \\
&\geq \frac{\eta}{2\sqrt{2}} \left\| v \right\|_2 - \frac{\eta}{32} \left\| u_P \right\|_2 \\
&\geq \frac{\eta}{2\sqrt{2}} \left\| v_P \right\|_2 - \frac{\eta}{32} \left\| u_P \right\|_2 \\
&\geq \frac{\eta}{2\sqrt{2}} \left\| u_P \right\|_2 - \frac{\eta}{2\sqrt{2}} \left\| (u - v)_P \right\|_2 - \frac{\eta}{32} \left\| u_P \right\|_2 \\
&\geq \frac{\eta}{8} \left\| u_P \right\|_2
\end{aligned}
$$

where the second and last inequalities apply Equation 7. We showed this inequality holds for all $a \in \mathbb{R}^m$ with probability at least $p$, which is the desired conclusion of the Lemma. This contradicts the initial assumption that the conclusion is false, proving the conclusion unconditionally. $\qquad \square$

### E.3 Failure of rectangular preconditioners

**Lemma E.7.** *Let $r := \dim \ker \Theta$. Then for any $S \in \mathbb{R}^{n \times s}$,*

$$\Pr_{i \in [n]} \left[ \left\| S^T e_i \right\|_1 \geq \frac{1}{\sqrt{\beta \epsilon n}} \right] > \frac{r - 2m}{n}$$

*where $\beta = \beta_{\tilde{\Theta}^{-1}, S, k, m}$.*

*Proof.* Suppose not. Then $\left\| S^T e_i \right\| < 1/\sqrt{\beta \epsilon n}$ for at least $n + 2m - r$ choices of $i \in [n]$. Pick some $\beta' > \beta$ which is sufficiently close to $\beta$ that $I = \{i \in [n] : \left\| S^T e_i \right\| < 1/\sqrt{\beta' \epsilon n}\}$ also satisfies $|I| \geq n + 2m - r$. Let $V = \operatorname{span}\{e_i : i \in I\}$. Then $\dim V = |I| \geq n + 2m - r$. Define $W = V \cap \ker \Theta$. Then $\dim W \geq \dim V - (n - r) \geq 2m$. Moreover, for any $w \in W$, we have

$$
\begin{aligned}
w^T \tilde{\Theta}^{-1} w &\geq \epsilon^{-1} \left\| \operatorname{Proj}_{\ker \Theta} w \right\|_2^2 \\
&= \epsilon^{-1} \left\| w \right\|_2^2
\end{aligned}
$$

whereas

$$\left\| S^T w \right\|_1 \leq \sum_{i \in I} |w_i| \cdot \left\| S^T e_i \right\|_1 \leq \frac{1}{\sqrt{\beta' \epsilon n}} \left\| w \right\|_1 \leq \frac{\left\| w \right\|_2}{\sqrt{\beta' \epsilon}}.$$

As a consequence, $w^T \tilde{\Theta}^{-1} w \geq \beta' \left\| S^T w \right\|_1^2$. Therefore $W \subseteq W_{\tilde{\Theta}^{-1}, S, \beta'}$, contradicting the definition of $\beta_{\tilde{\Theta}^{-1}, S, k, m}$. $\qquad \square$

**Lemma E.8.** *There are constants $c, c_m > 0$ so that the following holds. Let $M \in \mathbb{R}^{n-r \times n}$ and $\epsilon > 0$. Define $\Theta = M^T M$ and $\tilde{\Theta} = \Theta + \epsilon I$. Let $\lambda$ be the smallest nonzero eigenvalue of $\Theta$. Let $k, m, s, \alpha, \tau, \eta, b, b' > 0$ and suppose that $M$ satisfies $(b, b', \eta, \tau)$-erasure-robustness, and $\ker(M)$ is $(1/(12\sqrt{n}), 1/2\sqrt{n}, b')$-quantitatively-dense. Suppose $r > 2m$, $m \geq c_m \log n$, $k > (n/\tau) \log(sn)$*

*and $\epsilon < c\lambda/n^2$. Let $\mathcal{D}_k$ be the distribution of $k$-sparse signals with uniformly random support in $\mathbb{R}^n$, and Gaussian entries on the support. For any preconditioner $S \in \mathbb{R}^{n \times s}$ satisfying*

$$\Pr_{i \in [n-r]}\left[M_i^T \tilde{\Theta}^{-1} M_i < \alpha \left\|S^T M_i\right\|_1^2\right] \leq \frac{b}{n-r} \tag{10}$$

*and*

$$128 n^{7/2} k \eta \left\|M\right\|_F \sqrt{\frac{\epsilon \beta_{\tilde{\Theta}^{-1}, S, k, m}^{(1)}}{\alpha \lambda}} \leq 1, \tag{11}$$

*we have the following: with probability at least*

$$1 - 2/n - kb'/n - \exp(-k(r-2m)/n)$$

*over true signals drawn from $\mathcal{D}_k$, it holds that $S$-preconditioned Lasso fails with probability at least $1 - (4m)/(3r) - \exp(-\Omega(m))$ over independent samples $X_1, \ldots, X_m \sim N(0, \tilde{\Theta}^{-1})$.*

*Proof.* Let $S : n \times s$ be an arbitrary preconditioning matrix satisfying (10) and (11). By (10), we can apply Lemma D.2 to define sets $C, D$ as functions of $S$, with the following properties:

- For every $i \in D$, the column $v = (S^T)_i \in \mathbb{R}^n$ satisfies $\|v\|_0 \geq \tau$,

- The submatrix $S_{C^c D^c}$ satisfies

$$\sum_{j \in D^c} \|S_{C^c j}\|_2 \leq \frac{n^{3/2} \eta \|M\|_F}{\sqrt{\lambda \alpha}}$$

- $|C| \leq b'$.

Draw $w^* \sim \mathcal{D}_k$. Let $K = \text{supp}(w^*)$ and let $U = \text{supp}(S^T w^*)$. Recall that for every $i \in D$, the $i$th row of $S^T$ (the vector $(S^T)_i \in \mathbb{R}^n$) has at least $\tau$ nonzero entries. Therefore, for a particular choice of $i \in D$, the probability that $(S^T w^*)_i = 0$ is at most the probability that a uniformly random set of $k$ elements from $[n]$ misses all $\tau$ elements of the support of $(S^T)_i$, which is at most

$$(1 - \tau/n)(1 - \tau/(n-1)) \cdots (1 - \tau/(n-k)) \leq e^{-k\tau/n} \leq 1/(sn).$$

Hence by the union bound over all $i \in D$, recalling that $D \subset [s]$, we have that $D \subseteq U$ with probability at least $1 - 1/n$.

Similarly, because $|C| \leq b'$, with probability at least $1 - kb'/n$ it holds that $K \subseteq C^c$. By Lemma E.7, with probability at least $1 - (1 - (r-2m)/n)^k \geq 1 - e^{-k(r-2m)/n}$ it holds that $\|S_{j^*}\|_1 \geq 1/\sqrt{\beta \epsilon n}$ for some $j^* \in K$. Now conditioned on $K$, observe that $(w^*)_K$ has independent $N(0,1)$ entries. So by Lemma E.1,

$$\Pr[\|S^T w^*\|_1 \geq \|S_{j^*}\|_1 / n] \geq 1 - 1/n.$$

Thus, it follows that $\|S^T w^*\|_1 \geq 1/(n\sqrt{\beta \epsilon n})$ with probability at least $1 - 1/n - \exp(-k(r - 2m)/n)$ over $w^*$. Moreover $\|w^*\|_2 \leq 2\sqrt{k}$ with probability at least $1 - \exp(-\Omega(k))$. Assume for the rest of the proof that all of the above events (on $w^*$) occur: $D \subseteq U$, $K \subseteq C^c$, $\|S^T w^*\|_1 \geq 1/(n\sqrt{\beta \epsilon n})$, and $\|w^*\|_2 \leq 2\sqrt{k}$. We have shown that these together occur with probability at least $1 - 2/n - kb'/n - \exp(-k(r-2m)/n)$, and in the rest of the proof we show that under these events, $S$-preconditioned Lasso fails with probability at least $1 - (4m)/(3r) - \exp(-\Omega(m))$ over samples $X_1, \ldots, X_m$.

Let $z = \text{sign}(S^T w^*)$. Since $\text{supp}(w^*) = K \subseteq C^c$, we have by Cauchy-Schwarz that

$$\|(Sz)_{C^c}\|_2 \geq \frac{\langle w^*, Sz \rangle}{\|w^*\|_2} = \frac{\langle S^T w^*, z \rangle}{\|w^*\|_2} = \frac{\|S^T w^*\|_1}{\|w^*\|_2} \geq \frac{1}{2n\sqrt{\beta \epsilon n k}} \tag{12}$$

where the last inequality uses the above bounds on $\|S^T w^*\|_1$ and $\|w^*\|_2$. Define

$$d = \underset{x \in \ker X_{[m], C^c}}{\arg\min} \|x - (Sz)_{C^c}\|_2,$$

implicitly zero-extending $d$ from $C^c$ to $[n]$. Then $Xd = X_{[m],C^c}d_{C^c} = 0$ by construction, and moreover

$$\left\|(S^T d)_{U^c}\right\|_1 \leq \left\|(S^T d)_{D^c}\right\|_1 \qquad (D \subseteq U)$$

$$= \sum_{j \in D^c} |\langle (S^T)_{jC^c}, d_{C^c} \rangle| \qquad (\mathrm{supp}(d) \subseteq C^c)$$

$$\leq \|d\|_2 \sum_{j \in D^c} \|S_{C^c j}\|_2 \qquad \text{(Cauchy-Schwarz)}$$

$$\leq \|d\|_2 \cdot \frac{n^{3/2}\eta \|M\|_F}{\sqrt{\lambda\alpha}}.$$

On the other hand,

$$\|d\|_2^2 = \langle S^T d, z \rangle = \left\|\mathrm{Proj}_{\ker X_{[m],C^c}}(Sz)_{C^c}\right\|_2^2.$$

We now apply Lemma E.6 to vector $Sz$ and set $C^c$, using that $\ker(M)$ is $(1/(12\sqrt{n}), 1/(2\sqrt{n}), b')$-quantitatively dense in conjunction with the bounds $\epsilon \leq c\lambda/n^2$, $m \geq c_m \log n$, $r > 2m$, and $|C| \leq b'$. By this lemma and by (12), with probability at least $1 - (4m)/(3r) - \exp(-\Omega(m))$, we can lower bound the norm of the projection to get

$$\|d\|_2 \geq \frac{1}{64\sqrt{n}} \|(Sz)_{C^c}\|_2 \geq \frac{1}{128n^2\sqrt{\beta\epsilon k}}.$$

As a result, so long as $n^{3/2}\eta \|M\|_F /\sqrt{\lambda\alpha} < 1/(128n^2\sqrt{\beta\epsilon k})$, which holds by (11), we have that $\left\|(S^T d)_{U^c}\right\|_1 < \langle S^T d, z \rangle$ so the preconditioned Lasso fails on $w^*$. $\qquad \square$

**Theorem E.9.** *There are constants $c, c_m > 0$ so that the following holds. Let $M \in \mathbb{R}^{n-r \times n}$, and define $\Theta = M^T M$. Let $k, k', m, \tau, b, b', \eta, t > 0$. Let $\lambda$ be the smallest non-zero eigenvalue of $\Theta$. Suppose that $M$ satisfies $(b, b', \eta, \tau)$-erasure-robustness, and that $\ker(M)$ is $(1/(12\sqrt{n}), 1/(2\sqrt{n}), b')$-quantitatively-dense. Let $\epsilon > 0$ and define $\tilde{\Theta} = \Theta + \epsilon I$. Suppose that $544n^{15/2}k\eta \|M\|_F \sqrt{\epsilon/\lambda} \leq 1$, that $r > 2m$, that $m \geq c_m \log n$, and that $\epsilon < c\lambda/n^2$. And suppose that the rows of $M$ are $k'$-sparse. Then there is a distribution $\mathcal{D}$ over $\max(k't, k)$-sparse signals such that for any preconditioner $S \in \mathbb{R}^{n \times s}$ with $(n/\tau)\log(sn) < k$, with probability at least*

$$\frac{1}{2} \min \left( 1 - e^{-bt/(n-r)} - \frac{1}{n}, 1 - \frac{2}{n} - \frac{kb'}{n} - e^{-k(r-2m)/n} \right)$$

*over $w^* \sim \mathcal{D}$, the $S$-preconditioned Lasso fails to recover $w^*$ with probability at least $1 - (4m)/(3r) - \exp(-\Omega(m))$ over the samples $X_1, \ldots, X_m$.*

*Proof.* Let $\mathcal{D}_M$ be the signal distribution where we draw independent and uniformly random indices $R_1, \ldots, R_t \in [n-r]$ as well as independent $Z_1, \ldots, Z_t \sim \mathrm{Unif}([-1,1])$, and set the signal to be

$$w^* = \sum_{i=1}^t \frac{Z_i M_{R_i}}{\sqrt{M_{R_i}^T \tilde{\Theta}^{-1} M_{R_i}}}.$$

We define $\mathcal{D}$ to be the mixture which with probability $1/2$ draws a signal from $\mathcal{D}_M$, and with probability $1/2$ draws from the distribution $\mathcal{D}_k$ defined in Lemma E.8 (note that in the former case, the signal is $k't$-sparse, and in the latter case, it is $k$-sparse).

Pick any $S \in \mathbb{R}^{n \times s}$ with $(n/\tau)\log(sn) < k$. Define $\alpha = \beta^{(1)}_{\tilde{\Theta}^{-1},S,k,m}/(18n^4)$. We distinguish two cases.

**Case I: incompatible preconditioner.** On the one hand, suppose that

$$\Pr_{i \in [n-r]} \left[ M_i \tilde{\Theta}^{-1} M_i < \alpha \left\|S^T M_i\right\|_1^2 \right] \geq \frac{b}{n-r}.$$

Then with probability at least

$$\frac{1}{2} \left( 1 - (1 - b/(n-r))^t \right) \geq \frac{1}{2} \left( 1 - e^{-tb/(n-r)} \right)$$

over the row indices $R_1, \ldots, R_t$, there is some $R_i$ with

$$\left\| S^T M_{R_i} \right\|_1^2 > \frac{1}{\alpha} M_{R_i}^T \tilde{\Theta}^{-1} M_{R_i}.$$

In this event, we have

$$\max_{i \in [t]} \frac{\left\| S^T M_{R_i} \right\|_1}{\sqrt{M_{R_i}^T \tilde{\Theta}^{-1} M_{R_i}}} > \frac{1}{\sqrt{\alpha}},$$

so by Lemma E.1, it holds with probability at least $1 - 1/n$ over $Z_1, \ldots, Z_t$ that $\left\| S^T w^* \right\|_1 \geq \frac{1}{n\sqrt{\alpha}}$.
But by the triangle inequality, we have $\sqrt{(w^*)^T \tilde{\Theta}^{-1} w^*} \leq t \leq n$. Thus,

$$\Pr_{w^* \sim \mathcal{D}} \left[ \left\| S^T w^* \right\|_1^2 > \frac{1}{n^4 \alpha} (w^*)^T \tilde{\Theta}^{-1} w^* \right] \geq \frac{1}{2} \left( 1 - e^{-tb/(n-r)} \right) \left( 1 - \frac{1}{n} \right).$$

Moreover, for such $w^*$, by choice of $\alpha$ and by Theorem D.1, the $S$-preconditioned Lasso recovers $w^*$ with probability at most $\exp(-\Omega(m))$, from $m$ independent samples $X_1, \ldots, X_m \sim N(0, \tilde{\Theta}^{-1})$.

**Case II: compatible preconditioner.** On the other hand, suppose that

$$\Pr_{i \in [n-r]} \left[ M_i \tilde{\Theta}^{-1} M_i < \alpha \left\| S^T M_i \right\|_1^2 \right] \leq \frac{b}{n-r}.$$

This is precisely the condition (10) in Lemma E.8. Moreover, (11) is satisfied because we defined $\alpha$ so that $\beta_{\tilde{\Theta}^{-1}, S, k, m}^{(1)} / \alpha = 18n^4$, and because of the bound assumed in the theorem statement. Thus, we can invoke Lemma E.8, which states that for $w^* \sim \mathcal{D}_k$, with probability at least $1 - 2/n - kb'/n - \exp(-k(r - 2m)/n)$ over the signal, the $S$-preconditioned Lasso fails with the desired probability. Thus, the same holds for $w^* \sim \mathcal{D}$ up to a factor of 2. $\square$

Finally, we instantiate the above theorem with the matrix constructed in Theorem C.15.

**Theorem E.10.** *Let $n \in \mathbb{N}$ be sufficiently large. There is a matrix $\Sigma \in \mathbb{R}^{n \times n}$ with condition number $\mathrm{poly}(n)$ such that the following holds. Let $k \in \mathbb{N}$ with $k \geq \log^8 n$. There is a distribution $\mathcal{D}$ over $k \log^4 n$-sparse signals such that for any positive integer $m \leq n/5$ and any preconditioner $S \in \mathbb{R}^{n \times s}$ with $s < \exp(ck/\log^6 n - \log n)$, with probability at least $\frac{1}{2} - O(1/\log n)$ over $w^* \sim \mathcal{D}$, the $S$-preconditioned Lasso recovers $w^*$ with probability at most $8m/3n + \exp(-\Omega(m))$ from $m$ independent samples $X_1, \ldots, X_m \sim N(0, \Sigma)$ with noiseless responses $Y_i = \langle X_i, w^* \rangle$.*

*Proof.* Let $\Theta$ be the matrix guaranteed by Theorem C.15. We check the conditions of Theorem E.9. Clearly, $r := \dim \ker(\Theta) \geq n/2$. By claim (5), we have $\lambda = \Omega(n^{-5/2})$. By claims (3) and (2), we can take $b = n/(k \log n)$, $b' = 2n/(k \log n)$, $\eta = n^6 \log^8 n$, and $\tau = \Omega(n/\log^6 n)$. By claim (4), we have $\|M\|_F \leq O(\sqrt{n \log n})$. Thus, we can take $\epsilon = \Omega(n^{-22})$. By claim (1), the rows of $M$ are $k'$-sparse with $k' = O(\log^2 n)$. Let $t = k \log^2 n$.

So applying Theorem E.9, we get that there is a distribution $\mathcal{D}$ over $k \log^4 n$-sparse signals such that for any preconditioner $S \in \mathbb{R}^{n \times s}$ with $s < \exp(ck/\log^6 n - \log n)$, with probability at least

$$\frac{1}{2} \min \left( 1 - \frac{2}{n}, 1 - \frac{2}{n} - \frac{2}{\log n} - e^{-k(n/2 - 2m)/n} \right)$$

over $w^* \sim \mathcal{D}$, the $S$-preconditioned Lasso recovers $w^*$ with probability at most $8m/(3n) + \exp(-\Omega(m))$ from $m$ samples. So long as $m \leq n/5$, both terms in the minimum are $1 - O(1/\log n)$, yielding the claimed result. $\square$