# OpenReview forum: "Lower Bounds on Randomly Preconditioned Lasso via Robust Sparse Designs"
_NeurIPS.cc/2022/Conference — NeurIPS 2022 Accept_

### Official Review · Reviewer_dzk8 · 2022-07-08

**Rating:** 8
**Confidence:** 4
**Soundness:** 4 excellent
**Presentation:** 4 excellent
**Contribution:** 4 excellent

**Summary:**

This paper considers a sparse linear regression problem, where the covariate matrix is very-ill conditioned. This paper is motivated by the fact that there might be a computational-statistical tradeoff for this problem. That is, no polynomial-time algorithm exists which solves this problem in polynomial-time with an information theoretically (near-)optimal sampling rate.

This paper constructs a covariate matrix and a corresponding signal distribution such that any randomized $S$ preconditioned LASSO fails with high probability, if $S$ is a square matrix. In case that $S$ is a rectangular matrix contains a result of similar flavour. (However, in the latter scenario the paper only shows that any randomized $S$-preconditioned LASSO fails with probability $1/2$.) This gives first evidence that there might be a computational-statistical tradeoff in this problem.

Moreover, as a byproduct in the proof of the main result, this paper establishes a result about sparse linear regression with adversarial erasures.

**Questions:**

1. Many (all?!) other problems, which exhibit a computational-statistical tradeoff, like RIP certification or sparse PCA, can be reduced to the planted clique conjecture (at least in a certain sense). This seems not to be the case for this problem. It would be great if the authors could elaborate further on this.

2. LASSO is not the only algorithm one could apply to this problem. It would be great if the authors could comment on why they expect that properly modified variants of other algorithms like orthogonal matching pursuit also will not work in the low-sample regime.

3. In the abstract the authors write: "Sparse linear regression with ill-conditioned Gaussian random covariates is widely believed to exhibit a statistical/computational gap, but there is surprisingly little formal evidence for this belief." Are there any references where this possibility is discussed?

4. The paper uses the term LASSO to describe the optimisation problem (1). Would it not be more accurate to use the terminology "basis pursuit" instead?

**Limitations:**

Yes

**Strengths And Weaknesses:**

In contrast to other problems where these tradeoffs are suspected (k-Clique, sparse PCA,...) and reductions to the k-clique problem are known, sparse linear regression with ill-conditioned covariates is surprisingly poorly understood. This paper makes an important step towards better understanding this problem and identifying a potential computational-statistical tradeoff. Moreover, this paper is extremely well-written and it is a pleasure to read.

To conclude, I think this is a very strong paper which should published.

Typos:
l. 294: leas->leads

---

> ### Author Response · Authors · 2022-08-02
> **Author Response**
>
> We thank the reviewer for their time. To address their questions:
>
> 1. Indeed, ill-conditioned random-design sparse linear regression has no known reduction from planted clique. Of course, this is one of the primary motivations for our work. As to *why* this problem has no known reduction from planted clique, we can only point to a few intertwined partial answers.
>
>     First, as alluded to in the paper, planted clique reductions generally start with a reformulation as a simple testing problem between two very structured distributions which we believe is hard, e.g. test between $N(0,I)^{\otimes m}$ and $N(0, I + \theta uu^T)^{\otimes m}$ for sparse PCA. For SLR, there is no known analogue of this which is conjectured to be hard. This seems like the most fundamental obstacle.
>
>     Second, problems which reduce from planted clique generally seem to share some common structure: the hardness seems driven by a "signal-to-noise ratio'' (like $\theta$ in the sparse PCA example), which when varied transitions the problem from computationally tractable to computationally intractable to statistically impossible. It's not clear where to find this ratio in ill-conditioned SLR.
>
>     Note: there are many other problems with conjectured computational-statistical gaps which have no known reduction from planted clique. For example, the hardness of learning halfspaces agnostically is known in terms of SQ lower bounds and a reduction from a certain random constraint satisfaction problem (work of Daniely), but not from planted clique. See also parity with noise, learning halfspaces in the Massart noise model, nongaussian component analysis, k-community stochastic block model, mixture of  gaussians, ...
>
> 2. Good question; our general expectation is that since known guarantees for (standard) Orthogonal Matching Pursuit roughly  match known guarantees for (standard) Lasso, e.g. it works under incoherence/RIP/RE/etc., a preconditioner for Orthogonal Matching Pursuit should be "good'' roughly when it's "good'' for Lasso.  However, this is in no way rigorous, and it's possible that there is a way to precondition Orthogonal Matching Pursuit which could outperform preconditioned Lasso. This is one of the important directions of future study.
>
>     (Technical note: one case where OMP has stronger guarantees is under the Das and Kempe condition of weak submodularity, which is a bit weaker than the Lasso conditions. But this theory does not come with exact recovery guarantees, only approximate.)
>
> 3. Unfortunately, while the the performance of the Lasso on correlated/ill-conditioned data has been studied for a long time, we are not aware of a nice historical reference which explicitly posed this question/conjecture. We could call the statistical/computational gap for ill-conditioned Gaussian SLR a "folklore conjecture''. For a recent reference that discusses this possible gap, see e.g.
>
>     [Kelner et al., 2021] "On the Power of Preconditioning in Sparse Linear Regression" (e.g. their Discussion section)
>
>     There are also several papers which raise the possibility of a statistical/computational gap for the (very closely related/almost equivalent) problem of learning Gaussian Graphical Models, where again there is no formal evidence for the gap:
>
>     [Brennan et al., 2021] "Statistical Query Algorithms and Low-Degree Tests Are Almost Equivalent'' (see discussion on page 37)
>
>     [Kelner et al., 2020] "Learning Some Popular Gaussian Graphical Models without Condition Number Bounds" (see e.g. end of Section 1.1)
>
>     [Misra et al., 2020] "Information Theoretic Optimal Learning of Gaussian Graphical Models" (a paper on *inefficiently* learning ill-conditioned GGMs; they raise the computational question in the Conclusions section)
>
>     There are numerous papers on a (much smaller, roughly constant-factor) gap for isotropic Gaussian SLR (e.g. [Bandeira et al., 2022] "The Franz-Parisi Criterion"). But that problem seems unrelated to ours, where the conjectured sample complexity gap is nearly exponential.
>
>     There is also a closely related problem of learning sparse halfspaces --- this was posed by Feldman in a 2014 COLT Open Problem ("Open Problem: The Statistical Query Complexity of Learning Sparse Halfspaces'').
>
> 4. That is correct, Basis Pursuit would be more accurate terminology. Note that it is the limit of the Lasso program as the regularization parameter goes to zero.

---

> > ### Comment · Reviewer_dzk8 · 2022-08-07
> > **Response to Rebuttal**
> >
> > I want to thank the authors for their detailed answers to my questions. I stick to my original evaluation.

---

### Official Review · Reviewer_nP3f · 2022-07-10

**Rating:** 7
**Confidence:** 3
**Soundness:** 3 good
**Presentation:** 3 good
**Contribution:** 3 good

**Summary:**

The paper shows that the use of randomized preconditioners does not improve the performance of preconditioned lasso, as it is always possible to find data distributions (over sufficiently sparse signals) where the number of samples required by randomly preconditioned Lasso is linear. The analysis is tied to a result in compressed sensing that shows tolerance to adversarial erasure of randomized measurements that still allows partial recovery of the signal support, by exploiting the fact that binary random matrices are likely to provide adjacency matrices for expander graphs.


**Questions:**

Corollary 1.5 should be more explicit about what it means to identify "information theoretically" - the estimator in line 214 looks for the sparsest signal that meets a mismatch up to delta per observed measurement: how is this information-theoretic?

What does "morally" refer to in line 233? Is this discussion of preconditioners specific to random ones? Plenty of dense preconditioners can provide sparsity.

**Limitations:**

The compressed sensing result hinges on a non-practical recovery algorithm, hindering its applicability. It is also not fully clear how this aspect of the CS result impacts the preconditioner result - perhaps there could be a different recovery method that improves over the results here.

**Strengths And Weaknesses:**

The description in Section 2.1-2.2 is instructive.

It would be illuminating to have a discussion of the traits of the data distributions that nullify the possible improvement of randomized preconditioners. Perhaps this can inform whether randomized preconditioners may provide an advantage for the distribution at hand.

Although Theorems 1.2 and 1.3 easily track their counterparts in the supplement, it is less clear to see the relationship between Theorem 1.4 and its counterpart.

The acronym SLR should be defined.

---

> ### Author Response · Authors · 2022-08-02
> **Author Response**
>
> We thank the reviewer for their time. Addressing the reviewer's comments:
>
> 1. On traits of data distributions that nullify possible improvement of randomized preconditioners: at a high level, a deterministic preconditioner works when it can sparsely precondition the whole space; a randomized preconditioner can get away with preconditioning *most* of the space. See [Kelner et al., 2021] "On the Power of Preconditioning for Sparse Linear Regression'' (Appendix D) for an example along these lines where randomized preconditioning seems helpful.
>
> 2. In the formal statement of Theorem 1.4 (Theorem C.14) we use the terminology ``$(b,b',\eta,\tau)$-erasure-robust" which we formally defined in the preliminaries (line 634).
>
> 3. Thanks for catching this omission; we will add a mention that SLR stands for Sparse Linear Regression.
>
> Addressing the reviewer's questions:
>
> * By ''information-theoretic'' we simply mean to emphasize that the estimator we give is not computationally efficient. This is the standard terminology in the computational-statistical gaps literature.
>
> * When we say dense preconditioners "morally'' do not work, this is what we mean: if the signal is $e_i$ and column $i$ of the preconditioner $S^T$ is dense, then the preconditioned signal $S^T e_i$ is dense and because of this we do not expect our program based on $\ell_1$-penalization to recover the signal successfully. This intuition can be made rigorous when the preconditioner is square and invertible, in which case we are literally performing basis pursuit after changing basis by $S^T$: by a dimension counting argument, if $m$ samples are given, then the output of basis pursuit will be $m$-sparse with probability $1$, so a very dense signal will not be recovered. For rectangular preconditioners, we need to be more careful to rule out dense matrices and our techniques are more involved.
>
> [*Re. Limitations*]: the reviewer writes "The compressed sensing result hinges on a non-practical recovery algorithm... It is also not fully clear how this aspect of the CS result impacts the preconditioner result.'' We want to emphasize that the lower bound for preconditioned Lasso is unconditional; having an efficient algorithm for the erasure recovery problem is a very interesting open problem, but such a result would have zero implications for the lower bound. See the discussion at the end of Section 1.3 for more discussion of how information-theoretic erasure-robustness shows up as a technical ingredient in the lower bound proof.

---

### Official Review · Reviewer_S6CT · 2022-07-11

**Rating:** 5
**Confidence:** 2
**Soundness:** 2 fair
**Presentation:** 2 fair
**Contribution:** 3 good

**Summary:**

This paper investigated the problem of random-design sparse linear regression. Recent work shown that, for certain covariance matrices, the broad class of Preconditioned Lasso programs provably cannot succeed on polylogarithmically sparse signals with a sublinear number of samples. However, this lower bound only holds against deterministic preconditioners. For an appropriate covariance matrix, a single signal distribution is constructed on which an invertibly-preconditioned Lasso program fails with high probability, unless it receives a linear number of samples. Surprisingly, at the heart of our lower bound is a new robustness result in compressed sensing.

**Questions:**

1. What dose “statistical/computational gap” mean in the Abstract? And what is the “lower bound"? It is better to provide detailed explanations of these terms in this paper.
2. Please provide the reference for “Lasso” and its interpretation.
3. Is it possible to include numerical examples in this paper? This is helpful for readers to better understand the correctness and effectiveness of the results.

**Ethics Review Area:**

["I don’t know"]

**Strengths And Weaknesses:**

This manuscript studied recovering a sparse signal when a few measurements can be erased adversarially. Generally speaking, the paper is well-written, and enriched the existing results. However, the paper contains also several weaknesses:
1. The structure of the paper can be re-organized thus to improve the readability. So far, the paper contains only two sections. The flow of revealing the ideas and results is not very smooth and clear. The main contributions should be summarized in Section I. Moreover, (most of) the theoretical results established in the paper are presented in informal form. Then why not simply present them in the supplementary material and just cite them in the paper? These informal statements render it difficult to judge the results of the paper in a rigorous setup.
2. The paper is purely theoretical. The established computational and/or statistical results shall be compared against existing sparse recovery methods via substantial numerical simulations.

---

> ### Author Response · Authors · 2022-08-02
> **Author Response**
>
> We thank the reviewer for their time. Addressing the reviewer's comments:
>
> 1. Indeed, our main contribution is summarized in Section 1, on lines 66--68. The theorem statements in the introduction are written tersely (compared to the full statements in the supplementary) to avoid clutter, but they are entirely rigorous. We can certainly attempt to further clarify the statements in the final version.
>
> 2. Our main contribution is a proof that a whole class of algorithms cannot solve a particular statistical task. What numerical simulations would the reviewer have us run?
>
>     To see a numerical example where preconditioned Lasso works but standard Lasso does not work, see [Kelner et al., 2021] ''On the Power of Preconditioning in Sparse Linear Regression'' (Figure 1). But given that we are proving an impossibility result, it's not clear what empirical evidence would be apposite.
>
> Addressing the reviewer's questions:
>
> 1. See Line 40: a problem has a statistical/computational gap if $m_\text{est} \ll m_\text{alg}$, where $m_\text{est}$ is the number of samples needed to solve the problem information-theoretically, and $m_\text{alg}$ is the number of samples needed to solve the problem in polynomial time. The ''lower bound'' we prove is a lower bound on the number of samples needed by a particular class of algorithms (the preconditioned Lasso algorithms).
>
> 2. See Line 33, where we reference (Tibshirani, 1996: Regression shrinkage and selection via the lasso). The classical interpretation of Lasso is the following: ideally, we want to perform linear regression with $\ell_0$ regularization. But this is computationally intractable (for worst-case data), so we relax the $\ell_0$ penalty to a convex $\ell_1$ penalty. This yields the Lasso.
>
> 3. See response to comment 2.

---

### Official Review · Reviewer_ahcD · 2022-07-13

**Rating:** 6
**Confidence:** 3
**Soundness:** 3 good
**Presentation:** 3 good
**Contribution:** 3 good

**Summary:**

In this work, lower bounds on randomly preconditioned Lasso are provided. The authors construct a covariance matrix and a sparse signal distribution under which any randomly-preconditioned Lasso program with invertible preconditioners fails. The key technique is a new robustness result in compressed sensing, and the authors study recovering a sparse signal when a few measurements can be erased adversarially.

**Questions:**

I am wondering whether a matching upper bound can be achieved by computationally efficient algorithms.

**Limitations:**

Not applicable.

**Strengths And Weaknesses:**

Strengths: This paper is well-written. Both the main theoretical results (lower bounds on randomly preconditioned Lasso) and the key proof technique (erasure-robust sparse designs) are interesting to me. Although I have not checked the proofs in detail, the theoretical results seem to be reliable based on the technical overview.

Weaknesses: The authors should write down the full version (instead of informal statement) of the main theorems in the main text.

---

> ### Author Response · Authors · 2022-08-02
> **Author Response**
>
> We thank the reviewer for their time. To address their comments:
>
> * The ''informal statements'' in the introduction are written tersely to avoid clutter (i.e. using polylog instead of specifying the exact power), but they are entirely rigorous. However, we can certainly attempt to further clarify the statements in the final version.
>
> * Our lower bound is indeed matched by an upper bound (up to constants): our lower bounds show that (for our covariance matrix and signal distribution) no preconditioned Lasso algorithm can succeed with less than $n/7$ samples. On the other hand, with $n$ samples, the regression problem can always be solved by Gaussian elimination (and this is computationally efficient).
>
> (Note: in case the reviewer is asking about the adversarial compressed sensing result, as we stated in the text, making our estimator computationally efficient is an interesting open problem.)

---

### Meta-Review · Area_Chair_jWfR · 2022-08-20

**Recommendation:** Accept
**Confidence:** Certain

**Metareview:**

This paper studies the problem of sparse regression with ill-conditioned Gaussian covariates. Despite the simplicity of this problem formulation and the extensive studies of sparse linear regression, the potential existence of a statistical-computational gap for this problem has not been well understood. Taking a step towards understanding this problem, the authors provide theoretically rigorous evidence about the limitation of randomly preconditioned Lasso for this problem. The paper contains solid impossibility results, and hence I recommend acceptance.  Note that one reviewer has suggested ways to improve the structure and readability of the paper, which I hope the authors can address in the final paper; the paper would also benefit from having more substantial experiments.

**Award:**

No

---

### Decision · Program_Chairs · 2022-09-14

Accept